# HIV Infection: Shaping the Complex, Dynamic, and Interconnected Network of the Cytoskeleton

**DOI:** 10.3390/ijms241713104

**Published:** 2023-08-23

**Authors:** Romina Cabrera-Rodríguez, Silvia Pérez-Yanes, Iria Lorenzo-Sánchez, Rodrigo Trujillo-González, Judith Estévez-Herrera, Jonay García-Luis, Agustín Valenzuela-Fernández

**Affiliations:** 1Laboratorio de Inmunología Celular y Viral, Unidad de Farmacología, Sección de Medicina, Facultad de Ciencias de la Salud, Universidad de La Laguna (ULL), 38200 La Laguna, Spain; rcabrerr@ull.edu.es (R.C.-R.); sperezya@ull.edu.es (S.P.-Y.); ilorenzs@ull.edu.es (I.L.-S.); rotrujil@ull.edu.es (R.T.-G.); jesteveh@ull.edu.es (J.E.-H.); jgarcial@ull.edu.es (J.G.-L.); 2Analysis Department, Faculty of Mathematics, Universidad de La Laguna (ULL), 38200 La Laguna, Spain

**Keywords:** HIV-1 infection, HIV-1 Env complex, HIV-1 Pr55^Gag^, cytoskeleton dynamics, cytoskeleton adaptors and motor proteins, control of the infection, viremia and progression

## Abstract

HIV-1 has evolved a plethora of strategies to overcome the cytoskeletal barrier (i.e., actin and intermediate filaments (AFs and IFs) and microtubules (MTs)) to achieve the viral cycle. HIV-1 modifies cytoskeletal organization and dynamics by acting on associated adaptors and molecular motors to productively fuse, enter, and infect cells and then traffic to the cell surface, where virions assemble and are released to spread infection. The HIV-1 envelope (Env) initiates the cycle by binding to and signaling through its main cell surface receptors (CD4/CCR5/CXCR4) to shape the cytoskeleton for fusion pore formation, which permits viral core entry. Then, the HIV-1 capsid is transported to the nucleus associated with cytoskeleton tracks under the control of specific adaptors/molecular motors, as well as HIV-1 accessory proteins. Furthermore, HIV-1 drives the late stages of the viral cycle by regulating cytoskeleton dynamics to assure viral Pr55^Gag^ expression and transport to the cell surface, where it assembles and buds to mature infectious virions. In this review, we therefore analyze how HIV-1 generates a cell-permissive state to infection by regulating the cytoskeleton and associated factors. Likewise, we discuss the relevance of this knowledge to understand HIV-1 infection and pathogenesis in patients and to develop therapeutic strategies to battle HIV-1.

## 1. Introduction

Despite notable progress in human immunodeficiency virus (HIV)/acquired immunodeficiency syndrome (AIDS) prevention and treatment worldwide, this year United Nations (UN) experts warned that the epidemic remains a global concern requiring greater collaboration among countries, particularly in reaching the most vulnerable countries and populations [1]. The reasons why HIV is still currently a global threat range from social and economic points (since developing countries are hardly able to cover the needs of antiretroviral treatment (ART), which is even worse after the severe acute respiratory syndrome coronavirus 2 (SARS-CoV-2)/coronavirus disease 2019 (COVID-19) pandemic) to the biological and molecular features of the HIV virus itself [2]. In fact, small changes in the HIV type 1 (HIV-1) genome drive the escape of variants from the immune response, which makes the design of an efficient vaccine or therapeutic target against the infection highly difficult [3,4,5,6,7,8,9,10,11,12]. Furthermore, it is well-known that HIV-1 has developed different strategies to enter and infect the cell and use the cell machinery to ensure its survival, as is the case for viral factories that are “platforms” inside the cell where the virus is able to replicate and/or continue a part of its life cycle [13,14]. It is significant to keep in mind that the HIV-1 life cycle could usually be “split” into two main stages: early steps of the life cycle, which collect events from the HIV-1-mediated CD4 receptor and CXCR4 or CCR5 coreceptor interaction and signaling to the integration of the HIV-1 genome into the host cell (pre-integration phase) [15,16,17,18], and the late stages of the virus life cycle, which take into account HIV-1 genome replication until viral egress from the cell and its maturation (post-integration phase) [17,18,19,20,21,22] (Figure 1).

First, the HIV-1 envelope (Env) complex binds to the receptor CD4 and coreceptor CXCR4/CCR5, creating tension between the viral and cell membranes, leading to hemifusion of the membranes [23,24,25,26,27,28] that could evolve to the irreversible formation of the fusion pore, which occurs after the reorganization of the cytoskeleton and associated molecules and allows the liberation of the viral capsid into the cell cytoplasm. After entry, the viral core is transported towards the nucleus along microtubules while reverse transcription occurs. During this stage, it was thought that the core undergoes an uncoating process in which the capsid disassembles in the cytoplasm. However, several studies have indicated that the cone-shaped capsid remains intact during travel to the nucleus, which encases the genome and replication proteins. Therefore, the viral core serves as a reaction container for reverse transcription, a process that seems to be completed with uncoating at the nucleus [15,16,29,30,31,32,33,34,35,36,37,38,39,40,41,42,43,44]. The reverse transcription complex is actively transported into the nucleus through nuclear pore complexes, and upon entering the nucleus, HIV-1 integrase (IN) catalyzes the integration of retrotranscribed viral DNA into the host cell chromosomes [37,45,46,47,48,49,50,51]. All these steps involve the pre-integration phase [15,16,17,18] (Figure 1). From this point, the post-integration phase occurs: after the viral DNA is integrated into the host genome, viral mRNA is transcribed by the host RNA polymerase II machinery. Unspliced dimeric viral RNA is transported to the cell membrane by Gag (Pr55^Gag^), and this complex starts the assembly of nascent virions. During this process, the Pr55^Gag^ and Gag-Pol polyproteins polymerize around the viral RNA, gathering Env glycoproteins at the budding site, and virions bud off from the infected cell in an immature state. During or shortly after budding, the virion undergoes a maturation process in which the Pr55^Gag^ and Gag-Pol polyproteins are cleaved into separate mature proteins by the viral protease (PR) enzyme and it is at that moment when the mature virion is ready to infect a new target cell and reinitiate the viral life cycle [17,18,19,20,21,22] (Figure 1).

The complexity of the pathways implicated in the successful HIV-1 life cycle is outsized as the virus takes advantage of cell scaffolding and all the related molecules. In this regard, the cytoskeleton plays a very important role, as it performs functions associated with membrane dynamics such as cell migration, endocytosis, or trogocytosis but also during inner events such as autophagy or organelle displacement [52,53,54,55]. Moreover, studies related to membrane dynamics and cellular factors involved during early viral signaling and infection point out the relevance of understanding cytoskeletal modulation to tackle HIV-1 infection [52,56,57,58]. Indeed, it could be a very good prevention strategy to block HIV-1 signaling and entry into the target cell but also, in the case that the virus has integrated into the cell target genome, the strategy of impending viral egress has importance. For both strategies, the study of cytoskeletal dynamics is essential, revealing its role as a powerful tool to achieve the goal of fighting HIV-1 infection. Indeed, many works have pointed out that many viruses could use the endocytic and vesicle pathways during the early and late stages of infection to overcome the barrier that the cytoskeleton performs over them [52]. 

In this review, we gather the relevant information on HIV-1 cytoskeleton reorganization and dynamics using an approach from the three categories of the cell cytoskeleton: actin filaments (AFs), microtubules (MTs), and intermediate filaments (IFs) [59,60,61,62,63,64,65,66,67]. In addition, due to the high complexity of this cellular system and the biology of HIV-1, we treat those three components in the early and late steps of the HIV-1 life cycle to shed clarity on the whole process.

## 2. Modulation and Role of Cytoskeleton Activity during the Early Steps of the HIV-1 Viral Cycle (Pre-Integration Phase)

As reported before, we first collected the tracks from the pre-integration steps of the HIV-1 life cycle. It should be considered that in each phase of the cycle, there is an interaction of different categories of the cytoskeleton; for example, in pore fusion formation during viral entry, AFs and MTs as well as IFs have important roles, although each system is regulated in a different way. Briefly, AFs are the thinnest filaments in the cytoskeleton, are mostly composed of actin, and mediate the short-distance motility of cargos most frequently at the cell periphery; MTs are composed of polarized heteropolymeric chains of α/β tubulin with their minus-ends anchored at a perinuclear MT organizing center (MTOC) and their plus-end pointing toward the cell periphery, where they can interact with cortical actin and are responsible for the long-distance transport of cargos throughout the cell; and IFs help cells maintain their shape, bear tension, and provide cellular structural support (reviewed in [68]).

### 2.1. Actin Cytoskeleton and HIV-1 Infection

HIV-1 has developed different strategies to overcome the physical barrier imposed by cortical AFs to enter the cell and to exploit the actin network and its associated factors to facilitate early infection [68]. It is important to note that manipulation of host factors related to actin during HIV-1 infection could be either direct, when mediated by physical interaction with viral proteins, or indirect, when requiring upstream cellular factors [69]. In this regard, during the entry of HIV-1 into the target cell, there are multiple cell factors interfering in the recognition and specific binding of the gp120 subunit of the HIV-1 Env complex to CD4 and CXCR4/CCR5, after which different molecules signal to form the fusion pore [70,71,72,73,74,75]. It is thought that viral Env triggers an increase in the density of molecules for HIV-1 fusion in an actin-dependent manner that leads to the correct formation of the fusion pore [76]. 

For this purpose, there are so-called “cytoskeletal linkers”, such as the triad ezrin-radixin-moesin (ERM) actin-adaptor proteins, which have been reported to provide an inducible and reversible link between membrane-associated proteins and the actin cytoskeleton [77,78,79]. In fact, it has been reported that the first HIV-1 Env/CD4 interaction promotes the activation of moesin and thus triggers CD4/CXCR4 clustering and direct association [80]. Moesin phosphorylation in response to HIV-1 Env/CD4 binding is required for cell fusion, and this is related to the capacity of phosphorylated moesin to anchor and redistribute F-actin to a pole of the cell, where CD4 and CXCR4 reorganize, aggregate, and interact, thereby representing enriched CD4/CXCR4 sites for T-cell/virus contact, entry, and infection [80]. Notably, these hot sites or cell poles for infection are moesin/actin-driven pseudopods, where HIV-1 pore fusion formation and infection are favored [80] (Figure 2). In this sense, cell treatment with cytochalasin D, a drug that inhibits actin polymerization and induces depolymerization of AFs [81,82], negatively affects HIV-1 Env-mediated polarized cocapping of CD4/CXCR4 molecules and subsequent pseudopod formation [76]. Therefore, moesin appears to be a key factor to be activated by HIV-1 to assure efficient and productive infection of the cell (Figure 2).

It is plausible that other molecules able to interact with ERM proteins or microfilaments might regulate HIV-1 early infection to some extent, as has been suggested for the EWI motif-containing protein 2 (EWI-2)/α-actinin complex [83]. The silencing of these proteins seems to facilitate HIV-1 infection at the virological synapse (VS). Although it is known that EWI-2 associates with ezrin [84], an ERM actin-adaptor protein, and α-actinin binds to AFs [85,86,87], the mechanism of action of this complex on HIV-1 infection remains unknown. Moreover, reported data using an α-actinin-derived peptide (actin-binding site 1 peptide (ABS1p)) that blocks HIV-1 infection point to the importance of α-actinin-mediated dynamic cortical actin rearrangement for viral entry [88]. In this sense, published data about the role of the nucleotide-binding oligomerization domain (NOD)-like receptor (NLR) family pyrin domain-containing 3 (NLRP3) factor [89,90] in HIV-1 infection adds evidence to the importance of actin cytoskeleton dynamics for virus-mediated fusion and entry [91]. NLPR3 represses F-actin remodeling, an activity that could hinder HIV-1 entry. In fact, HIV-1 Env-mediated virus–cell interaction activates the P2Y2 purinergic receptor that recruits the E3 ubiquitin ligase Casitas B lymphoma (Cbl) protein to NLRP3, promoting its degradation, allowing actin cytoskeleton remodeling, and thus viral entry [91].

It is important to note that the F-actin-linker moesin also codistributes with constitutive components of clathrin-coated structures to regulate the formation of membrane-cargo-derived endosomes, as well as their internalization and trafficking [92]. Therefore, it appears that the actin-adaptor moesin, which is activated by HIV-1, drives membrane dynamics at the cell surface. Hence, this study showed that moesin knockdown by using specific short interfering RNA (siRNA) enhances the lateral movement of clathrin-coated structures and leads to their abnormal clustering. Likewise, similar results are obtained when using a dominant-negative moesin construct unable to bind to F-actin and a moesin construct unable to bind to the phosphotidylinositol 4,5-bisphosphate (PIP_2_) associated with fluid cargo membranes [92]. In fact, moesin knockdown provokes the accumulation of nascent and early endocytic clathrin-coated vesicles associated with the Ras-related protein in brain 5 (Rab5) small guanosine triphosphatase (GTPase), and carrying of the transferrin receptor (TfR) [92]. The altered trafficking of endocytic Rab5-clathrin-coated vesicles accounts for a TfR recycling defect that reduces cell surface expression of TfR and increases the amount of sequestered Tf ligand [92]. Altogether, these data represent a scenario in which actin reorganization and membrane dynamics interact through the HIV-1-associated moesin ERM factor.

In this regard, it was published that binding of the HIV-1 Env-gp120 protein to CD4 induces PIP_2_ production, which regulates plasma membrane fluidity [93,94,95], in permissive lymphocytes through the activation of phosphatidylinositol-4-phosphate 5-kinase (PI4P5-K) Iα [96]. The overexpression of wild-type PI4P5-K Iα increased HIV-1 Env-mediated PIP_2_ production and enhanced viral replication in primary lymphocytes and a CD4+ T-cell line, whereas PIP_2_ production and HIV-1 infection were both severely reduced in cells overexpressing a kinase-dead mutant D227A (D/A)-PI4P5-K Iα [96]. Likewise, HIV-1 infection is inhibited by the knockdown of endogenous expression of PI4P5-K Iα, indicating that PI4P5-K Iα-mediated PIP_2_ production is crucial for HIV-1 entry and infection in the early steps of the viral cycle [96].

In turn, it is conceivable that HIV-1 regulates actin cytoskeleton reorganization and dynamics for viral infection by acting on the PI4P5-K Iα/PIP_2_ axis to modulate cytoskeletal molecules such as gelsolin, profilin, filamin-A, and ERM proteins [97,98,99] (Figure 2). In fact, the moesin protein, like other ERM F-actin-linkers from the band 4.1 superfamily [100,101,102,103,104], presents the KK(X)n(K/R)K consensus binding site for PIP_2_ at the N-terminal part of the molecule [105]. The recognition of PIP_2_ by moesin is required for the conformational activation of the ERM [92,106,107] and combined K/N mutations of the Lys residues 253 and 254, and 262 and 263 of the actin-adaptor (i.e., the 4K/4N-moesin mutant [92]) entirely abrogate the interaction of the ERM with PIP_2_ [77,92,108], lacking CCV-membrane trafficking control [92] and redistributing to the cytoplasm [77,92,108]. A similar effect is achieved in ezrin by combining the double mutation of residues K63N and K64N with the double K253N and K254N mutation [108]. In this context, the PIP_2_ pool at the inner leaflet of the plasma membrane could either recruit these actin-related proteins to specific membrane domains or trigger conformational changes within these target proteins to be activated or inhibited. Hence, the PI4P5K I/PIP_2_ axis controls several biological processes related to AFs redistribution, such as cell polarity [77,80,109], cytokinesis [99], phagocytosis [110], cell migration, and the formation of membrane protrusions [111]. In this context, HIV-1 viral entry occurs at specific cell surface areas enriched in F-actin and viral receptors, such as ruffles, microvilli, and pseudopods [80,112,113,114], where plasma membrane regeneration is also needed, in an ADP ribosylation factor 6 (Arf6)/PIP_2_-dependent manner [14,115]. These F-actin structures are governed by cortical actin cytoskeleton dynamics, which in turn depend on the activity of actin-adaptors or actin-binding proteins. It is therefore conceivable that the PI4P5K Iα/PIP_2_ axis could regulate the formation of these structures and promote HIV-1 entry and infection (Figure 2).

Filamin-A is another actin-adaptor protein relevant to how HIV-1 modulates the cortical actin cytoskeleton to assure productive infection [109]. Thus, HIV-1-Env-mediated signals promote the link between cortical AFs and the CD4/CXCR4 HIV-1 receptors through the filamin-A molecule [109]. In addition, HIV-1 Env/CD4 signaling activates the GTPase RhoA, which inactivates cofilin (a depolymerizing/actin-severing protein [116,117,118]), thereby assuring the stability of cortical AFs [109], which are polymerized by the HIV-1 Env/CD4-mediated activation of cortical moesin [80]. Filamin-A anchors these cortical AFs where they accumulate, and through its association with HIV-1 receptors, it generates a cap of CD4 and CXCR4 [109]. Therefore, HIV-1 uses moesin and filamin-A to promote the aggregation of HIV-1 receptors in a pole of the cell, thereby regulating CD4/CXCR4-CCR5 diffusion at the cell surface in an actin cytoskeleton-dependent manner [80,109,114,119] to facilitate the first events of the infection process (Figure 2).

One important question emerges to be solved to understand the way HIV-1 promotes cortical AFs dynamics to reorganize, move, and generate a “hot-viral entry” pseudopod in CD4+ T cells. The actin-severin gelsolin could help one to find the answer and to understand this biological process. In fact, gelsolin appears to be essential for efficient and productive HIV-1 infection by regulating the size of preformed AFs by cleaving them into short filaments that allow the actin cytoskeleton to reorganize together with CD4/CXCR4-CCR5 aggregation at pseudopods, where the virus enters and infects CD4+ T cells [114]. Notably, the imbalance of the steady-state levels of expression of gelsolin, by overexpressing functional gelsolin or by specific siRNA knockdown of endogenous gelsolin, results in aberrant or defective levels and sizes of AFs, respectively. Hence, HIV-1 is unable to reorganize aberrant AFs in the absence of gelsolin, negatively affecting pseudopod formation and CD4/CXCR4-CCR5 redistribution and aggregation, thereby avoiding viral infection [114]. In contrast, an excess of active gelsolin abrogates the formation of AFs by HIV-1 Env/CD4 interaction, impeding all the above-described dynamic events to infect target cells. Therefore, the level of expression of the gelsolin enzyme could restrict HIV-1 infection of CD4+ lymphocytes at a prefusion step (Figure 2).

Regarding the regulation of actin cytoskeleton dynamics at the postfusion steps of the HIV-1 entry and infection process, a study highlighted the importance of cofilin in disturbing the barrier represented by cortical AFs, which are driven by the above-presented factors and events, in a CD4-dependent manner [120]. After fusion, the anchored cortical actin cytoskeleton in T cells could hinder viral capsid uptake, and HIV-1 uses G_αi_-dependent signaling from the CXCR4 coreceptor to activate a cellular F-actin-depolymerizing/severing factor, cofilin, to overcome this AFs’ restriction in resting cells [120]. HIV-1 Env/CXCR4-mediated cofilin activation is important for a post-entry process that leads to viral nuclear localization through depolymerizing AFs. Therefore, inhibition of HIV-1-mediated actin cytoskeleton rearrangement by blocking cofilin action markedly diminishes latent viral infection of resting T cells, and conversely, induction of active cofilin greatly facilitates it [120]. In another study, latency established in resting T CD4+ cells was associated with rapid dephosphorylation of cofilin and changes in filamentous actin after exposure to ligands for CCR7 (CCL19), CXCR3 (CXCL9 and CXCL10), and CCR6 (CCL20), which was not observed in unactivated CD4+ T cells, indicating complex signaling from the cytoskeleton and chemokine molecules [121]. Furthermore, at this postfusion step, phosphorylated moesin by the action of the initial HIV-1 Env/CD4 interaction, which links with cortical AFs through its phosphorylated C-terminal domain of the molecule, becomes inactive by dephosphorylation, relaxing the tension established between the plasma membrane and AFs, thus allowing the entry of the viral capsid (i.e., HIV-1 genome) into the cell [80] (Figure 2).

Likewise, it has been observed in T cells derived from HIV-1-infected patients, with or without ART, significantly lower levels of hyperactive or phosphorylated cofilin compared with those cells from healthy controls, with cofilin hyperactivation also associated with poor CD4+ T-cell recovery following ART [122]. These results suggest an HIV-1-mediated systemic dysregulation of T-cell motility that cannot be repaired solely by ART. Furthermore, it has been reported that stimulation of blood CD4+ T cells with an anti-human α4β7 integrin antibody (Ab) triggers the cofilin pathway, partially restoring T-cell motility in vitro [122]. However, severe T-cell motility defects caused by high degrees of cofilin hyperactivation were not repairable by the anti-integrin Ab, demonstrating a mechanistic hindrance to restore immune functions in vivo. In summary, cofilin is a key molecule that may need to be therapeutically targeted early for immune control of viremia [122]. Notably, in 2008, a pilot study observed phosphorylated cofilin in resting CD4+ T cells purified from the peripheral blood of HIV-1-infected patients. The study reported that the resting T cells from infected patients carried significantly higher levels of active cofilin, suggesting that these resting cells had been primed in vivo in cofilin activity to facilitate HIV-1 infection. It is important to consider that the procedure used to purify blood CD4+ T cells could imply the centrifugation of these cells. In this matter, cell infection by centrifugal inoculation (spinoculation) seems to trigger dynamic actin and cofilin activity, probably resulting from cellular responses to centrifugal stress, so this actin activity also leads to the upregulation of the HIV-1 CD4 and CXCR4 receptors, enhancing viral binding and entry [123]. It was also observed that an actin inhibitor, jasplakinolide, diminishes spin-mediated enhancement and that siRNA knockdown of LIM domain kinase 1 (LIMK1; a cofilin kinase) decreases the enhancement [123]. These results suggest that spin-mediated enhancement cannot be explained simply by a virus-concentrating effect; rather, it is coupled with spin-induced cytoskeletal dynamics that promote receptor mobilization, viral entry, and post-entry processes [123]. In addition, it is important to note that HIV-1-mediated aberrant activation of cofilin may also lead to abnormalities in T-cell migration and activation that could contribute to viral pathogenesis [124].

Furthermore, the main above-presented data have been integrated and quantitatively analyzed by using a mathematical model to calculate how different molecules act during HIV-1 infection [125]. The results obtained indicate that moesin activation is induced by virus signaling, while filamin-A is mobilized after AFs are formed, driving receptor capping. The disaggregation of AFs from the cap is facilitated by cofilin. Cofilin is inactivated by HIV-1 signaling in activated lymphocytes, while in resting lymphocytes, another signal is required to activate cofilin in the later stages to accelerate the decay of aggregated AFs as a restriction factor for viral entry [125]. Furthermore, stopping the activation signaling of moesin is sufficient to liberate the AFs from the cap, allowing HIV-1 capsid entry and infection. Finally, gelsolin must cleave AFs in short filaments to allow actin cytoskeleton reorganization during early HIV-1 infection steps [125].

Altogether, these results indicate that in this scenario where HIV-1 promotes CD4-mediated signals to polymerize and reorganize the cortical actin cytoskeleton, filamin-A and gelsolin/cofilin, together with moesin, have emerged as key cellular factors driving this actin cytoskeleton dance, in turn representing a barrier for HIV-1 infection (see summary in Table 1).

The cortical actin cytoskeleton is therefore the immediate structure intercepting the virus after membrane fusion, and for that reason, there is a possibility that actin-associated factors or actin activity itself may be a direct force for uncoating. The uncoating process is still a great topic of discussion, as researchers have three different points of view depending on whether the capsid is totally uncoated in the cytoplasm immediately during entry through the fusion pore [42,44,126,127,128,129,130,131,132], partially uncoated in the cytoplasm until the pre-integration complex (PIC) reaches the nuclear pore [40,133,134,135], or uncoating occurs entirely at the nucleus [136,137,138,139]. However, as indicated above, recent studies indicated that the HIV-1 capsid remains intact during travel to the nuclear envelope, which encases the genome and replication proteins. Therefore, the viral core serves as a reaction container for reverse transcription, a process that seems to be completed with uncoating at the nucleus [15,16,29,30,31,32,33,34,35,36,37]. In fact, it has been proven that the actin cytoskeleton and AFs-MTs cross-linking through their subunit proteins are implicated in the regulation of uncoating and reverse transcription, as the factors Diaphanous-related formins (DRFs), Dia1, and Dia2, which control both actin nucleation and MT stabilization [140,141], facilitate the intracellular motility of the viral capsid, and Dia2 binds viral capsids to mediate the uncoating process and retrotranscription [142]. The uncoating process of the viral capsid could be detrimental to infection if it exposes the viral genome before entry into the nucleus of cells [15,136,139,143,144,145], as reported for nonstable HIV-1 viral capsids [143,146,147,148], which are the main antiviral function of the classical tripartite motif containing 5 alpha (TRIM5a) restriction factor [149,150], cyclophilin A [151], and Trim5-cyclophilin A fusion protein (TRIMCyp) [152].

Nevertheless, there are also viral factors that play a role during viral infection in the pre-integration stages. The HIV-1 negative factor (Nef) is a small myristoylated protein that could affect early postfusion steps, leading to increased infectivity [153,154]. In this sense, HIV-1 Nef in the PIC is known to directly interact with actin, suggesting possible anchorage of PIC onto the cortical actin cytoskeleton for efficient reverse transcription, enhancing viral infectivity. This enhancement has been attributed to a positive effect of Nef at post-entry steps, such as uncoating or reverse transcription. HIV-1 Nef has been known to interact with the HIV-1 core and requires cellular cofactors for enhancement, as Nef-defective virions carry normal levels of endogenous reverse transcriptase activity in CD4+ T lymphocytes and macrophages [153,155,156,157,158,159,160]. Notably, HIV-1 lacking Nef is infection defective, and the infectivity of Nef-deficient virions could be complemented by drugs disrupting the cortical actin cytoskeleton [161] or by pseudotyping virions with the vesicular stomatitis virus glycoprotein (VSV-G), which fuses viral particles in low pH endocytic vesicles.

Moreover, the phosphorylated HIV-1 Pr55^Gag^ matrix protein (MA) is mainly associated with the HIV-1 PIC and with the actin cytoskeleton during the early stages of HIV-1 infection in T-cell lines, determining the localization of reverse transcription to actin microfilaments that was mediated by the interaction of a reverse transcription complex component (Pr55^Gag^-MA) with actin but not vimentin (IFs) or tubulin (MTs). Furthermore, productive HIV-1 infection seems to be impaired when actin-depolymerizing agents are added to target cells before infection but not when they are added after infection [162].

Finally, it is important to indicate that functional primary HIV-1 Envs have been reported to efficiently reorganize cortical AFs to generate the pseudopod fusion and entry zone to productively infect permissive cells, and these HIV-1 Envs, isolated from viruses of infected patients, are associated with the progressor clinical phenotype [163]. In contrast, deficient HIV-1 Envs that reorganize cortical AFs and infect are associated with viruses isolated from long-term nonprogressor elite controller (LTNP-EC) patients [163].

A summary of the main data discussed in this section is presented in Table 1.

### 2.2. Microtubules and HIV-1 Infection

The tubulin cytoskeleton is a key component of the cellular scaffold that HIV-1 modulates to accomplish pore fusion formation and infection. In the regulation of MT dynamics during HIV-1 early infection, the acetylation-deacetylation balance is known to be decisive and carried out by histone deacetylase 6 (HDAC6) [71,74,75,163,164,165,166,167,168], an enzyme mainly located in the cytoplasm that regulates the deacetylation of the α-tubulin subunit in MTs [168,169,170,171,172]. The acetylation of the α-tubulin subunit at the Lys^40^ residue, which is located in the inner or luminal part of the MTs [173,174,175], could be considered a posttransductional modification (PTM) marker for MT stabilization [172,174,176,177,178,179,180,181,182,183,184,185]. Notably, it has been observed that the specific binding of the gp120 protein of the HIV-1 Env complex to CD4 at the cell surface of permissive cells increased the level of acetylation of MTs in the α-tubulin subunit [56,70,71,163,164,165]. In fact, overexpression of deacetylase-active HDAC6 inhibited the α-tubulin acetylation of MTs by HIV-1, preventing HIV-1 Env-mediated membrane fusion and infection without affecting the expression and distribution of HIV-1 receptors (i.e., CD4, CCR5, and CXCR4), while the knockdown of HDAC6 expression or inhibition of its tubulin-deacetylase activity strongly enhanced HIV-1 Env-mediated cell-to-cell fusion (syncytia formation (virus-mediated fused giant multinucleated cells)) and HIV-1 infection of primary CD4+ T cells [164] (Figure 3).

HIV-1-triggered acetylation of MTs renders MTs hydrophobic, which could favor the close association of the cortical tubulin cytoskeleton with the plasma membrane directly or through molecules linked with the membrane, thereby creating an anchoring or tension zone to favor pore fusion formation and infection near the virus–cell contact and signaling regions. In this regard, it has been reported that acetylated MTs associate with the membrane by interacting with the transport protein Na^+^, K^+^-ATPase [186,187,188,189], as well as with other ATPases [190]. Therefore, because the fusion peptide of the HIV-1 gp41 viral protein is thought to insert into target membranes [191,192,193,194] to mediate lipid exchange between HIV-1 and plasma membrane lipid bilayers, producing a fusion pore at the viral-cell contact region, the anti-HIV-1 effect exerted by the tubulin-deacetylase HDAC6 could be due to the impediment to establish an HIV-1 gp41/plasma membrane/acetylated MT “tension zone” during the early steps of the infection process. Likewise, HDAC6 inhibits viral entry and infection, precisely impeding pore fusion formation [71,164,166]. Thus, HDAC6 impairs pore fusion formation from the six-helix bundle (6HB) intermediate that is promoted by the class I viral gp41 fusion protein [23,28,195,196,197,198,199,200] and required for the fusion of the virus–cell membranes [25,27,28,201]. In fact, the conformational folding of the 6HB intermediate is irreversibly complete after pore fusion formation [24].

Moreover, several studies that have characterized the functionality of the HIV-1 Env complex isolated from viruses of patients with extreme clinical phenotypes shed light on the events that occur in early infection, where HDAC6 plays an important role acting as a tubulin-deacetylase enzyme. In a cluster of HIV-1 LTNP-EC patients, it has been shown that these individuals have been infected by virus bearing an inefficient HIV-1 Env complex to bind to and to signal through CD4 [163]. These LTNP-EC HIV-1 Envs could not efficiently trigger the acetylation of MTs, thereby they were unable to overcome the barrier that endogenous HDAC6 tubulin-deacetylase represents in permissive CD4+ cells. These LTNP-EC HIV-1 Envs are therefore ineffective in ensuring productive CD4-dependent viral fusion, transfer, and infection [163], directly correlating these deficient viral signals and functions to the LTNP-EC clinical outcome [163]. These functional data of HIV-1 Envs from viruses of a cluster of LTNP-EC individuals, infected with a similar transmitted/founder (T/F) virus and without clinical progression, for more than 30 years [163,202], have been further confirmed in Env complexes of nonclustered viruses from LTNP-EC individuals [74]. Therefore, LTNP-EC individuals present viruses with inefficient HIV-1 Envs to induce acetylation of MTs, a PTM required for pore fusion formation and productive infection. On the other hand, HIV-1 Envs isolated from the virus of viremic patients (progressors, rapid progressors (RPs) and viremic nonprogressors (VNPs)) have been reported to be able to overcome the anti-HIV-1 action of the HDAC6 restriction factor, thereby promoting acetylation of MTs, as their Env complexes are fully functional and cytopathic [70] (all these data and concepts are briefly commented on in Figure 3).

Therefore, the extension of the acetylation-PTM of MTs, triggered by viral Env, conditions the efficiency of HIV-1 early infection. Notably, the antiviral control exerted by HDAC6 renders the regulation of the expression of this tubulin-deacetylase a key goal to protect against HIV-1 infection. In this sense, the transactive response DNA-binding protein (TARDBP; also known as transactive response DNA-binding protein 43 kDa (TDP-43)) [203,204,205,206] regulates the stability of the mRNA and therefore conditions the protein levels of the antiviral HDAC6 enzyme [57,71,207]. Regarding this, a recent study reported the functional implication of TDP-43 in determining cell permissivity to HIV-1 infection by modulating the level of expression of HDAC6 mRNA and enzyme, and therefore the acetylation level of MTs [71]. The overexpression of TDP-43 negatively affected HIV-1 Env fusion and infection capacity by stabilizing tubulin-deacetylase HDAC6, and decreasing acetylated MTs, independently of Env tropism. Consistently, silencing endogenous TDP-43 significantly decreased HDAC6 levels, increasing the acetylation of the α-tubulin of MTs, which favors the fusogenic and infection activities of HIV-1 Env [71]. Likewise, this anti-HIV-1 HDAC6/TDP-43 axis regulates the infection activity of HIV-1 Envs of the virus isolated from VNP and RP patients down to the levels of the inefficient HIV-1 Envs from viruses of LTNP-EC individuals (Figure 3).

An interesting observation, which demonstrates the relevance of the acetylation PTM of MTs for productive HIV-1 infection, was the fact that silencing of the endogenous TDP-43/HDAC6 axis significantly favored the infectivity of primary Envs of viruses isolated from VNP and RP HIV-1 patients and strongly increased the infection of those from LTNP-EC that in control conditions are inefficient for infection [71]. Hence, TDP-43 shapes cell permissivity to HIV-1 infection, affecting viral Env fusion and infection capacities by altering HDAC6 levels and associated tubulin-deacetylase anti-HIV-1 activity. Altogether, these data provide a clear demonstration that defective viral features observed in LTNP-EC individuals could be modulated by the TDP-43/HDAC6 axis that maintains MTs in a deacetylated state [71]. It is thus interesting to seek potential defective functional mutations of TDP-43/HDAC6 in some LTNP-EC individuals who have lost natural viremic control.

Once the HIV-1 core enters the cell, it appears that MTs serve the viral capsid in traveling “long distances” to the nuclear membrane [40,145,208,209] to accomplish viral RNA+ retrotranscription and subsequent integration of the double-stranded viral DNA (dsDNA), named complementary or copy DNA (cDNA), into the host genome [37,45,46,47,48,49,50,51]. Recent evidence suggests that capsid uncoating occurs at the nucleus when cDNA synthesis is completed [15,210,211,212,213]. Intracellular HIV-1 capsids require stable acetylated MTs to enter cells [71,163,164,166,214], as well as MT-associated molecular motors, such as dynein [40] and dynein adapter protein bicaudal D2 (BICD2) [215,216], which control viral core travel speed and transport direction, in order to traffic to the nuclear pore complex (NPC) (Figure 3), where nucleoporins, particularly nucleoporin Nup153, mediate HIV-1 core translocation into the nucleus [210]. Notably, the acetylation of MTs leads to the recruitment of molecular motors to these stabilized MTs, increasing the trafficking flux of cargos along microtubular tracks [217,218,219,220], which occurs by overcoming HDAC6-tubulin-deacetylase activity [221,222,223,224,225,226,227]. In this sense, as referenced before, DRFs Dia1 and Dia2 control both actin nucleation and MT stabilization and remodeling [140,141,228], promoting stabilization of MTs during infection to facilitate perinuclear trafficking of HIV-1 cores [142]. The DRFs are related to the EB1 (end-binding protein 1)-kif4 (kinesin family member 4) pathway to stabilize MTs [229,230], which in turn are involved in HIV-1/EB1-mediated MT acetylation [214] and stabilization by core/kif4 interaction [214]. Likewise, fasciculation and elongation protein zeta-1 (FEZ1) has been proposed to mediate the association of the HIV-1 core with the MT motor kinesin-1 to reach the nucleus [231], a process regulated by an MT affinity-regulating kinase 2 (MARK2) [232] that also allows HIV-1 to turn the kinesin-1 motor on or off during the bidirectional transport [40,145] and it is required to achieve effective net viral core perinuclear transport [232].

From the point of view of viral factors that take advantage of MT scaffolding, it is known that the +TIP proteins (plus tip-tracking proteins) rapidly induce MT stabilization and facilitate the delivery of viral particles to the nucleus by their interaction with the MA protein of HIV-1 [142]. For example, the capsid binds Dia1/2 to couple uncoating and stable MT-based viral movement, or the cytoplasmic linker protein (CLIP)-associated protein 2 (CLASP2), which is a key regulator of cortical capture and MT stabilization that is an EB1-associated +TIPs protein that binds HIV-1 capsids and regulates early infection [214,233,234]. In a good way for its life cycle, the HIV-1 core mimics MT regulators in a very similar manner [235]. Notably, it is plausible that the alteration of the HIV-1 core interaction with these cytoskeleton-associated motors could provoke abnormal virus trafficking [236] and facilitate the recognition of the viral material by antiviral cellular sensors (i.e., TRIM5α or TLRs (toll-like receptors)) [143,149,150,152,237,238,239,240,241,242], thus activating immune cellular responses to inhibit viral infection/replication [36,243,244,245,246,247,248], as reported for unstable HIV-1 cores [143,144,236,249,250]. These facts reinforce the relevance of MT stability [71,163,164,166,214] and nuclear core uncoating for viral infection (comprehensively reviewed in [37]).

Furthermore, it is interesting to note that primary HIV-1 Envs of viremic and progressing HIV-1 patients stabilize acetylated MTs [70,163], correlating this function with their cytopathic effects exerted on bystander cells by promoting late autophagy [56,70]. In this sense, it has been reported that hyperacetylation of MTs is required for autophagy stimulation under different biological events [251,252,253], and there are several autophagic proteins that directly or indirectly interact with MTs during the autophagic process [254,255]. Light chain 3 (LC3), a mammalian homologue of autophagy-related protein 8 (ATG8), has been shown to associate with MTs indirectly through an interaction with microtubule-associated proteins (MAPs) MAP1A, MAP1B, and MAP1 small form (MAP1S; originally named chromosome 19 open reading frame 5 (C19ORF5)) [256,257,258,259], but LC3 might also interact directly with tubulin through its N-terminal domain [260]. Some ATG proteins, such as Unc-51-like autophagy activating kinase 1 (ULK1), Beclin 1 (BECN1; the mammalian orthologue of Atg6 in yeast), WD repeat domain, phosphoinositide interacting 1 (WIPI1) protein, ATG5, or ATG12, that are involved in the early steps of autophagosome formation are enriched in the dynamic MT fraction, suggesting that the dynamic subset of MTs supports the assembly of preautophagosomal structures [261,262,263,264]. Many researchers point out that MTs do not have any active mechanism in this process, but what is very likely is that, at least, MTs are scaffolds for the autophagy pathway during HIV-1 infection. The mammalian target of the rapamycin (mTOR) factor binds to the surface of lysosomes, and its activity is controlled by lysosome localization, which is organized by MTs [265,266]. Nevertheless, autophagy is involved in HIV-1 infection in a cell type-dependent manner (comprehensively reviewed in [70,267,268]) and could even be usurped by HIV-1 to facilitate viral protein processing and virion assembly [70]. Altogether, these data suggest that in viremic and progressing HIV-1 patients, HIV-1 Envs, exposed both in virions and on infected cell surfaces, establish long-lasting contacts with bystander, noninfected permissive cells via CD4 or CXCR4 and CCR5 receptors, thereby promoting persistent acetylation of MTs to lead to fatal, nonregulated, late autophagy, which could reduce the number of immune competent cells [70,267,268,269,270,271,272,273] and be associated with immune senescence [274,275,276,277,278,279,280,281], even in ART-treated patients [282,283,284,285].

A summary of the main data discussed in this section is presented in Table 2.

### 2.3. Intermediate Filaments and HIV-1 Infection

The role of intermediate filaments (IFs) during early HIV-1 infection is less well understood compared to the functional implication demonstrated for AFs and MTs at the fusion, entry, and viral infection steps. There is recent evidence that indicates that the structural type III IF vimentin protein [286,287,288,289,290,291], one of the most studied IFs, could be involved in the first HIV-1/cell contacts by attaching the gp120 subunit of the viral Env complex [292]. More data are needed to understand the functional role of this HIV-1 Env/vimentin binding in the context of the viral cycle and whether there is a positive or negative association between HIV-1 infection and pathogenesis. Moreover, vimentin has been reported to be associated with several HIV-1 proteins, such as virion infectivity factor (Vif) and viral PR. Data about these HIV-1 proteins and vimentin are discussed in Section 3, since PR and Vif are key HIV proteins for viral particle maturation and infectivity [57,75,293,294,295,296,297,298,299,300,301,302,303,304,305,306,307,308,309,310,311], and the experimental models used in the analyzed vimentin/HIV-1 studies do not allow us to determine the potential involvement of vimentin in early or later stages of the virus replication cycle.

## 3. The Role of the Cytoskeleton during Late Steps of the HIV-1 Life Cycle (Post-Integration Phase)

### 3.1. Actin Filaments

In HIV-1 infection, cell-to-cell viral transmission at VS has been well documented and is mainly responsible for the high decrease in T CD4+ lymphocytes during the acute phase but especially during the AIDS phase of infection, thus having clinical importance [312]. This manner of infection involves different pathways but includes events in which an infected cell makes contact with a noninfected cell, transmitting the virus in different ways: during VS (transmission or transfer) and by cytoskeletal structures, such as filopodia or nanotubes [14,312,313]. Hence, during viral egress, AFs stand out as scaffolds for the virus to bud. The plasma membrane is able to form structures such as filopodia or nanotubes for cell-to-cell viral transmission, as filopodia extend through receptor-mediated mechanisms from uninfected cells towards the infected cell, and a similar retroviral surfing has been described over nanotubes that also transfer viral proteins inside [14,52,314,315,316,317,318,319,320,321]. In this case, receptor specificity seems to play a minor role, as these actin-driven structures seem to extend from HIV-1-infected cells to target cells in the absence of receptor–Env interactions [14,22,52,316,317,318,319,320]. In DCs, by budding at the tip of filopodia, HIV-1 can tether several neighboring CD4+ T cells, leading to viral transfer and infection of the targeted T cells [322]. Nanopores are structures formed between cells, approximately 100 nm in length, and depend on microfilament structures [323] that, in the context of HIV-1 VS, are formed in the membrane of the target cell in which the virus “surfs” through that structure to infect and in an irrespective manner of receptor–HIV-1 Env interaction [314]. Infection-mediated nanotubes could be established between T CD4+ lymphocytes (infected and noninfected pairs) or between T CD4+ lymphocytes and dendritic cells (DCs) [314]. In this matter, during the infection of DCs, HIV-1 (which enters the cell using the lectin DC-SIGN) mediates the activation of a GTPase and the remodeling of the actin cytoskeleton to promote filopodia extension that allows virus transmission to neighboring CD4+ T cells [314,322].

The ERM actin-adaptors also have relevance in HIV-1 production [324], since these proteins connect cortical F-actin with integral and peripheral membrane proteins they are incorporated into virions, interacting with cellular components of the virological presynapse. Phosphorylation activates ezrin, which specifically accumulates at the HIV-1 presynapse in T-cell lines and primary CD4+ lymphocytes [324]. Moreover, while cells did not tolerate a complete knockdown of ezrin, even a modest reduction in ezrin expression of approximately 50% in HIV-1-producing cells led to the release of particles with impaired infectivity. Furthermore, when cocultured with uninfected target cells, ezrin-knockdown producer cells displayed reduced accumulation of the tetraspanin CD81 at the VS and formed syncytia. Such an outcome is likely not optimal for virus dissemination, as evidenced by the fact that, in vivo, only relatively few infected cells form syncytia. Thus, ezrin likely helps secure efficient virus spread not only by enhancing virion infectivity but also by preventing excessive membrane fusion at the VS [324]. Likewise, moesin regulates HIV-1 Env/CD4-mediated pore fusion formation in a cell-to-cell VS model, where moesin phosphorylation and its actin/plasma membrane-anchoring activity are required [80].

Recently, the factor EWI-2, a protein that was previously shown to associate with ezrin and tetraspanins, has been identified as a host factor that contributes to the inhibition of HIV-1 Env-mediated cell–cell fusion [325]. It has been observed that EWI-2 accumulates at the presynaptic terminal where it contributes to the fusion-preventing activities of the other viral and cellular components, and EWI-2 was also downregulated upon HIV-1 infection, most likely by the accessory viral protein U (Vpu) [325]. In this sense, the expression levels of EWI-2 and CD81 molecules are restored on the surface of HIV-1 Env-driven syncytia [325], where they act as fusion inhibitors, thereby providing novel insight into how deathly syncytia (i.e., collapsing by apoptosis [326,327,328]) might be prevented from fusing indefinitely. This ‘syncytial apoptosis’ is frequently detected during HIV-1 infection in vitro and in vivo in tissues from HIV-1-infected patients [327]. Hence, and in contrast to potential EWI-2/CD81 inhibitors [325], it has been proposed that the induction of selective death of HIV-1-elicited syncytia might lead to the elimination of viral reservoirs. Thus, specific drugs or strategies aligned with this concept would constitute a therapeutic complement to current ARTs [328].

One key factor that could associate cortical AFs with the cell surface and appears to regulate late stages of the HIV-1 infection cycle is the interferon (IFN)-induced type II membrane glycoprotein tetherin, also known as bone marrow stromal cell antigen 2 (BST2), HM1.24 or CD317 [329,330]. Tetherin is a GPI anchor protein localized in lipid raft microdomains that can link rafts with the underlying cortical actin cytoskeleton [331] and acts as a direct physical tether for virions, linking the HIV-1 Env complex to the plasma membrane and preventing the final “escape” of viruses from the infected producer cell [332,333,334,335,336,337]. Tetherin thus severely limits the spread of cell-free viruses and reduces the cell-to-cell spread of HIV-1 to a lesser degree [338,339]. HIV-1 uses its Vpu protein to counteract the BST2/tetherin barrier [340,341]. Vpu localizes predominantly to the trans-Golgi network (TGN) and recycling endosomes [342,343]. Thus, Vpu antagonizes tetherin in a post-ER compartment [344], downregulating BST2/tetherin from the cell-surface [341] by recruiting a βTRCP2-SCF-Cullin1 ubiquitin ligase complex [345,346,347,348,349,350].

P-selectin glycoprotein ligand 1 (PSGL-1) is another important factor recently related to HIV-1 replication that could restrict actin dynamics and arrest viral Env at the plasma membrane, resulting in virions with poor Env incorporation and reduced infectivity [351]. In fact, PSGL-1 is a dimeric mucin-like glycoprotein that binds to P-, E-, and L-selectins and plays a role in leukocyte rolling on endothelial surfaces prior to transmigration [352,353,354,355,356,357], being identified as an IFN-γ-induced restriction factor in CD4+ T cells using a genome-wide proteomic screen [358]. In the context of HIV-1 infection, PSGL-1 seems to be recruited to the sites where HIV-1 particles assemble [359]. Likewise, it seems that PSGL-1 inhibits reverse transcription in target cells and diminishes the infectivity of released virions. This inhibitory effect could be antagonized by the HIV-1 Vpu protein through the ubiquitination and proteasomal degradation of PSGL-1. Therefore, PSGL-1 could be incorporated into HIV-1 virions, exerting a negative effect on HIV-1 transmission, not only by inhibiting Env processing and incorporation but also by directly inhibiting virion attachment to target cells [360].

A recent study concluded that HIV-1 molds AFs cortical nodes in areas of high positive membrane curvature within Arp2/3-dependent F-actin filopodia and lamellipodia structures, which enables HIV-1 to bud at the cell edge [361]. This work is supported by live cell imaging and focused ion beam scanning electron microscopy (FIB-SEM), which permit the observation of F-actin structures that exhibit strong positive curvature where HIV-1 buds [361]. Likewise, virion proteomics, gene silencing, and viral mutagenesis supported not only the functional involvement of Arp2/3 but also the involvement of the Rac1- and Cdc42-IQGAP1 pathways in driving F-actin regulation and membrane curvature, thus allowing HIV-1 to egress. HIV-1 also activates the Cdc42-Arp2/3 filopodial pathway for release, requiring the scaffolding protein IQGAP1 (a Rac1 and Cdc42 binding partner) [362,363], a process that similarly occurs during cell-to-cell viral spreading [361].

The HIV-1 structural Pr55^Gag^ polyprotein could be coimmunoprecipitated with actin but not tubulin, where a direct physical interaction between HIV-1 Pr55^Gag^ and F-actin via the nucleocapsid (NC) domain of Pr55^Gag^ has been reported [364]. Likewise, it is known that there is a physical and tight association between HIV-1 Pr55^Gag^ and actin both in fixed and adherent cell lines [365]. In addition, several studies have observed a high association of F-actin with HIV-1 budding sites, as some researchers have observed NC-dependent actin structures that emanate from HIV-1 buds during viral assembly and disappear upon viral release [366]. Additionally, the presence of AFs near budding sites was later confirmed and proven to have direct physical contact with nascent viral particles [367,368] (Figure 4). However, some observations argue against a functional role of these microfilaments in viral budding [368,369], but since these studies did not measure the actin content in cell-free viral supernatants, a role for actin in HIV-1 egress could not be ruled out. Therefore, given the importance of viral RNA binding to the NC region of the Pr55^Gag^ polyprotein during the viral life cycle, functional characterization of the NC-actin association and how this association is influenced by the viral RNA+ genome should be further addressed to understand the role played by AFs in HIV-1 Pr55^Gag^ packaging and virion release.

A summary of the main data discussed in this section is presented in Table 3.

### 3.2. Microtubules

It is thought that HIV-1 release and cell-to-cell transfer rely on intact and stable acetylated MTs [164,166,370,371], such as through VS, where during transfer, the MTOC in HIV-1-infected T cells, but not in target cells, polarizes towards the VS [370,371,372]. Notably, in infected peripheral blood lymphocytes (PBLs), the stabilization of MTs by chemical inhibition of the tubulin-deacetylase HDAC6 increases HIV-1 viral production and replication for several days postinfection [164]. This fact indicates the importance of acetylated stable MTs for the late stages of the HIV-1 life cycle and points to the regulatory role of HDAC6 in HIV-1 replication.

In this sense, the tubulin-deacetylase HDAC6 appears to be crucial in the control of HIV-1 production and the infection capacity of nascent viral particles [57,58,75,164]. HDAC6 directly interacts with A3G (apolipoprotein B mRNA-editing enzymatic polypeptide-like 3G or APOBEC3G) to undergo cellular codistribution along MTs and the cytoplasm in an A3G/HDAC6 complex [75]. HDAC6 competes for Vif-mediated A3G degradation, also accounting for the steady-state A3G expression level. In fact, HDAC6 directly interacts with and promotes Vif autophagic clearance due to its C-terminal BUZ domain, a process requiring the tubulin-deacetylase activity of HDAC6 [75]. HDAC6 degrades Vif but does not affect core-binding factor β (CBF-β), a Vif-associated partner reported to be key for Vif-mediated A3G degradation. Thus, HDAC6 antagonizes the proviral activity of the Vif/CBF-β-associated complex by targeting Vif and stabilizing A3G. Finally, in cells producing virions, a good correlation was observed between the ability of HDAC6 to degrade Vif and restore A3G expression, suggesting that HDAC6 controls the amount of Vif incorporated into nascent virions and the ability of HIV-1 particles to be infectious [75]. The tubulin-deacetylase HDAC6 also promotes the autophagic degradation of the viral polyprotein Pr55^Gag^ to inhibit HIV-1 production [58]. The HIV-1 Nef factor counteracts this antiviral activity of HDAC6 by inducing its cellular clearance and subsequently stabilizing Pr55^Gag^ and Vif viral proteins. HIV-1 Nef interacts with HDAC6, colocalizing on MTs and neutralizing HDAC6 by degradation in an acidic/endosomal-lysosomal pathway [58]. HIV-1 Nef therefore assures Pr55^Gag^ location and aggregation at the plasma membrane, promoting virus production and enhancing the infectivity of viral particles by the stabilization and uptake of Vif [58]. Moreover, a recent study showed that the overexpression of TDP-43 in virus-producing cells stabilizes HDAC6 (at the mRNA and protein levels) and triggers the autophagic clearance of HIV-1 Pr55^Gag^ and Vif proteins [57]. These events inhibit viral particle production and impair virion infectiveness, resulting in a reduction in the amount of Pr55^Gag^ and Vif proteins incorporated into virions. [57]. A nuclear localization signal (NLS)-TDP-43 mutant is not able to control HIV-1 viral production and infection. Likewise, specific TDP-43 knockdown reduces HDAC6 expression (i.e., mRNA and protein) and increases the expression levels of HIV-1 Vif and Pr55^Gag^ proteins, thereby stabilizing MTs by α-tubulin acetylation [57]. Thus, TDP-43 silencing favors virion production and enhances virus infectious capacity, thereby increasing the amount of Vif and Pr55^Gag^ proteins incorporated into virions. Notably, there is a direct relationship between the content of Vif and Pr55^Gag^ proteins in virions and their infection capacity [57] (Figure 4).

Furthermore, the concomitant PTM of MTs by acetylation produced after silencing the HDAC6/TDP-43 axis could stabilize Pr55^Gag^, facilitating its transport to the plasma membrane. In this sense, it has been reported that the Pr55^Gag^ polyprotein colocalizes, early after expression, with viral RNA through the cis-acting packaging element, psi (Ψ), in the perinuclear region and centrioles. Ψ+ RNA and Pr55^Gag^ are subsequently transported to the plasma membrane of the cell [365]. Moreover, HIV-1 Pr55^Gag^ uses MT-dependent cellular machinery to traffic to the cell surface in a manner dependent on the suppressor of cytokine signaling 1 (SOCS1) [373], where stable MTs are key for SOCS1/Pr55^Gag^ transport [373]. SOCS1 is a negative regulator of cytokine signaling [374] and interacts with the MA and NC regions of Pr55^Gag^ through its central SH2 domain to facilitate Pr55^Gag^ intracellular trafficking and stability, thereby regulating the late stages of the virus replication pathway [375] (Figure 4).

In contrast, some MTs-associated molecules, such as the IQGAP family of scaffold proteins, which also regulate the actin cytoskeleton [328,376,377,378], have been reported to exert an inhibitory effect on HIV-1 budding and maturation by targeting Pr55^Gag^ [379]. IQGAP1 has been shown to interact with the budding of enveloped viruses, as is the case for HIV-1 [380,381,382]. Overexpression of IQGAP1 inhibits HIV-1 budding, while depletion of IQGAP1 enhances HIV-1 particle release [379]. In fact, IQGAP1 directly interacts with both the NC and p6 regions of the structural Pr55^Gag^ viral protein, altering and impairing its distribution at the plasma membrane, thereby being responsible for the inhibition of viral release [379] (Figure 4).

An example of HIV-1 viral factors that play an important role in the late stages of the viral cycle and interact with tubulin cytoskeleton are the accessory Tat (transactivator of transcription) and viral protein R (Vpr) proteins [383,384].

It has been observed that HIV-1 Tat interacts with monomeric tubulin and MTs. This Tat–MT association does not appear to alter or be required for Tat secretion or uptake. However, the association of Tat with MTs induces apoptosis, as the combination of Tat and MT leads to the alteration of MT dynamics and activation of a mitochondria-dependent apoptotic pathway [383]. In addition, Bim facilitates Tat-induced apoptosis, since Bim is a proapoptotic Bcl-2 relative and a transducer of death signals initiated by perturbation of MT dynamics [383]. Therefore, the proapoptotic Tat–MT interplay could account, in part, for the immune cell dysfunctions and pathogenesis associated with HIV-1 infection. HIV-1 Vpr is able to interact with MT-associated factors, such as EB1, p150^Glued^, and dynein heavy chain [384], and alters the MT plus end localization of EB1 and p150^Glued^, negatively affecting the centripetal movement of phagosomes and their maturation [384]. Moreover, Vpr appears to target MT/dynein-dependent endocytic trafficking in HIV-1-infected macrophages, which deeply alters phagolysosome biogenesis [384]. It is plausible that this Vpr-provoked cell dysfunction could be toxic for infected cells by acting on caspases [385,386,387], and mitochondria-associated factors (reviewed in [388,389,390]) that could be activated after phagolysosome disruption [391].

A summary of the main data discussed in this section is presented in Table 4.

### 3.3. Intermediate Filaments

The distribution of vimentin IFs around the nucleus and their extension throughout the cytoplasm provide a scaffold for cellular components (reviewed in [392]) and allow vimentin to play an essential role in cellular cargo transportation [393,394,395,396,397,398,399,400,401,402], as well as in the trafficking of viral components to the location of cell assembly and egress (reviewed in [403,404]). It is interesting to note that vimentin IFs are distributed in cells by MTs-dependent dynamic transport [405,406,407,408,409], regulated by AFs and related kinases [410], which, in turn, is required to maintain the cytoskeleton network [411,412]. Vimentin IFs modulate the replication, assembly, and egress of viruses in the host due to their known function of regulating endosomal trafficking via Rab7a and Polo-like kinase 1 (Plk1). Rab7a, which is ubiquitously present in early and late endosomes [413,414], interacts with insoluble and soluble vimentin [415,416]. Rab7a interacts directly with vimentin, and this interaction modulates vimentin phosphorylation and assembly [415]. Rab7a-depleted cells have an abundance of insoluble vimentin IFs and defective endosomal trafficking [417]. Phosphorylation of vimentin IFs at Ser^459^ by Polo-like kinase 1 (Plk1) inhibits endolytic fusion during mitosis [398]. Altogether, these interactions demonstrate that vimentin is a critical regulator of late endocytic trafficking and egress of viral particles [418,419,420,421].

In the context of HIV-1, the knockdown of the vimentin protein has been associated with a reduction in the HIV-1 p24 capsid (CA) protein (derived from Pr55^Gag^) level with concomitant impairment of viral replication [13]. Likewise, the intracellular administration of a peptide that modifies vimentin IF distribution similarly inhibits HIV-1 replication [13]. Moreover, the interplay of vimentin IFs with the Mac-2-binding protein (M2BP [422]), a protein that is secreted in HIV-1-infected patients and correlates with viremia progression or control [423,424,425], mediates the association of vimentin IFs with HIV-1 Pr55^Gag^ [426]. Thus, vimentin/M2BP interplay negatively affects the trafficking of HIV-1 Pr55^Gag^ towards the plasma membrane, whereas the absence of M2BP favors the cell surface localization of the HIV-1 Pr55^Gag^ polyprotein [426,427].

It is interesting to note that some clinical studies with HIV-1 patients suggest vimentin as a potential marker for ART treatment efficiency, since in HIV-1-positive peripheral blood mononuclear cells (PBMCs) isolated from patients, vimentin was verified to be related to ART treatment and viremia control [428]. Although there are certain limitations of this study, the reported data could be of interest. For example, the protein–protein interaction analysis was database dependent, and limited clinical samples were obtained to verify the expression level of vimentin in plasma under different experimental conditions [428]. Moreover, several reports point out that some viruses could rearrange vimentin filaments (reviewed in [403]), and in the case of HIV-1, some published data support the possibility that vimentin may serve as a substrate for HIV-1 proteins within infected cells [429]. In this sense, the IFs vimentin, desmin, and glial fibrillary acidic proteins are cleaved in vitro by the HIV-1 PR enzyme [429]. Thus, microsequencing analysis showed that HIV-1 PR cleaved both human and murine vimentin. Microinjection of HIV-1 PR into cultured human fibroblasts resulted in a 9-fold increase in the percentage of cells with an altered and abnormal distribution of vimentin IFs [429]. The ability of the HIV-1 PR to liberate amino-terminal peptides from vimentin, representing one of the best substrates of the viral PR, may be important to achieve the replication cycle because PR-derived vimentin fragments disrupt IFs organization [430,431]. In the case of HIV-1 Vif, this accessory protein is found in the cytoplasm and in the perinuclear area colocalizing with vimentin, thereby disrupting the vimentin network [432]. In contrast, Vif presents a diffuse pattern of distribution in the cytoplasm, nuclear membrane and nucleus when the vimentin network is chemically blocked [432]. 

Most commonly, when the vimentin network is perturbed, their IFs collapse into a clump with a juxtanuclear localization. Similarly, amino-terminal polypeptides of vimentin are responsible for changes in nuclear architecture associated with HIV-1 PR activity in tissue culture cells isolated from infected individuals, and condensation or degeneration of nuclear chromatin has been described in cells from a variety of tissues [433]. Furthermore, infection with HIV-1 includes an increase in the incidence of cancer, particularly lymphomas. Attempts at identifying an additional agent responsible for non-Hodgkin’s lymphoma in a large HIV-1-infected cohort have failed [434], raising the possibility that HIV-1 itself is directly responsible. The ability of the HIV-1 PR to liberate amino-terminal peptides from vimentin may interfere with host cell gene expression [429,435] and may be important to achieve the replication cycle because PR-derived vimentin fragments disrupt IF organization [430,431]. In the case of HIV-1 Vif, this accessory protein is found in the perinuclear area colocalizing with vimentin, thereby disrupting the vimentin network [432]. Altogether, these data indicate that the relevance of the HIV-1 Vif/vimentin IF interplay for HIV-1 replication remains to be elucidated.

## 4. Discussion and Future Perspectives

The cytoskeleton represents the main barrier that HIV-1 encounters and must overcome during infection to complete its viral cycle. Notably, the three main cytoskeleton structures (i.e., AFs, MTs, and IFs) could play either a positive or a negative role in the HIV-1 life cycle. For example, AFs reorganized by HIV-1, particularly through the activation of ERM actin-adaptors, are able to form filopodia and pseudopod “hot-viral entry structures” that help HIV-1 in the first steps of infection (i.e., pore fusion formation and productive cell entry) [78,80]. The aggregation of cortical actin in the regions where HIV-1 enters cells appears to be also required for the uptake of the virus by immune DCs, since these actin structures govern the clustering of the sialic acid binding Ig-like lectin 1 (SIGLEC1 or CD169) that captures viral particles and leads to virus sequestration within so-called virus containing compartments [436]. On the other hand, actin-severing factors such as gelsolin or cofilin could redistribute and unbalance the quantity and shape of F-actin filaments to impede or promote viral infection [114,120]. Gelsolin tailors the size of the virus-induced AFs, allowing their redistribution to a pole of the CD4+ T cell, where HIV-1 signals to form the pseudopod, to fuse and infect [114]. This regulation of the cortical actin cytoskeleton and associated factors is key to generating a cluster zone where CD4/CCR5-CXCR4 receptors aggregate and directly interact in a moesin/filamin-A-dependent manner [80,109,114,163]. HIV-1 Env drives these events during the first virus–cell contact to increase the probability of Env-CD4/CCR5-CXCR4 encounters, favor membrane fluidity, and create an HIV-1 gp41/plasma membrane/AF adaptor tension area required for virus–cell membrane lipid exchange that ultimately forms the fusion pore to mediate virus core entry [80]. This last step is under the control of moesin inactivation by dephosphorylation and the AFs-cleavage activity of cofilin that generates an AFs-free zone while the HIV-1 capsid enters to initiate the viral infection process [80,120].

In this scenario, HIV-1 Env concomitantly modifies and stabilizes the tubulin cytoskeleton by promoting hydrophobic acetylated MTs at the virus–cell contact zones to favor HIV-1 gp41 fusion activity [164] and postfusion viral core recruitment to the MTs tracks and associated molecular motors to travel to the NCP, where the core anchors nucleoporins and translocates into the nucleus. The tubulin-deacetylase HDAC6 acts against HIV-1 Env/CD4-mediated MT acetylation, impairing pore fusion formation, viral entry, and infection [164]. The TARDP/TDP-43 factor controls the expression level of HDAC6, and therefore, the HDAC6/TDP-43 axis regulates cell permissiveness to HIV-1 infection [71]. The knockdown of this pair of antiviral proteins enhances cell infection even by inefficient HIV-1 Env from LTNP-EC individuals [71]. It is important to note that efficient HIV-1 Envs to regulate the cytoskeleton during the first virus–cell contacts are also involved in cytotoxic events, such as apoptotic syncytia formation and late autophagy that eliminate bystander cells [70,327]. Therefore, the efficiency and extension of the first HIV-1 Env signals could condition the progression rate of the infection, virus variability, and associated pathogenesis, which may be related to the natural control of the infection and disease in LTNP individuals (i.e., viremic LTNPs or ECs) [56,165,328].

Once the HIV-1 core enters the cell, MTs-associated factors that could regulate tubulin cytoskeleton stability and/or function as molecular motors condition the efficiency of the next step of the HIV-1 infection cycle. Dia1/Dia2 formins are MTs stabilizers that remodel MTs by acting on the EB1-kif4 pathway, which facilitates perinuclear trafficking of HIV-1 capsids [142,214,229,230]. The direction and speed of the viral core travel to the nucleus are under the control of the MTs-related dynein and BICD2 motors, as well as by MTs-associated factors that mediate the link between HIV-1 capsid and tubulin tracks during trafficking to the nuclear envelope, such as the FEZ1/kinesin 1 or CLASP2/EB1 complex [40,145,231,232].

In the case of IFs, cell surface vimentin appears to act during the first HIV-1/cell contacts by binding to the gp120 subunit of the HIV-1 Env complex [292]. It is important to consider that plasma membrane inclusions of vimentin have been related to its dysregulation and the induction of several malignant properties, as described in colon cancer metastasis [437], lung cancer progression [438], breast cancer [439], and lymphoma [440,441,442]. Thus, considering immune cells, gp120/vimentin association at the cell surface might occur both in T-cell and B-cell lymphoma, with vimentin predominantly expressed at the cell surface of B-cell lymphoma [440,442]. Therefore, whether the gp120/vimentin interaction aggregates viral particles on the cell surface “vimentin lattice” either as a kind of viral bioavailability reservoir for subsequent infection together with CD4/CCR5-CXCR4 receptors or as a decoy to neutralize viral particles is totally unknown.

All these fine-tuned regulated mechanisms are essential to restrict HIV-1 early infection when HIV-1 Envs are deficient to reorganize the cytoskeleton to create a permissive cell state, as in the case of HIV-1 virus of LTNP-EC individuals [71,74,163,165]. In contrast, HIV-1 Envs from viremic and progressing HIV-1 patients productively modulate the actin cytoskeleton, their associated factors, and cell structures, and stabilize acetylated MTs, overcoming the endogenous HDAC6/TDP-43 antiviral axis to fuse, enter, and infect permissive cells [70,71,74,163,165]. Therefore, determining the exact role of each actin/tubulin-associated protein involved in virus-triggered signaling and cell structures could help to explain the mechanism underlying HIV-1 infection, latency, and reservoir stabilization, as well as virus persistence and pathogenesis, therefore helping to understand HIV-1 patients’ clinical outcomes [74].

HIV-1 also exploits the cell cytoskeleton to acquire the late stages of the viral cycle by mainly stabilizing its structural Pr55^Gag^ polyprotein and trafficking it to the plasma membrane, where the virus reorganizes or takes advantage of cortical cytoskeleton dynamics to curve the plasma membrane to permit Pr55^Gag^ assembly and a viral budding edge [361,362,363,364,365,366,367,368]. In turn, Pr55^Gag^ together with other HIV-1 accessory proteins interact and stabilize the cytoskeleton to favor viral egress. Notably, further Pr55^Gag^ processing in immature virions will release its different subunits to shape the morphology of the infective virion.

The antiviral activity of the tubulin-deacetylase HDAC6 or the HDAC6/TDP-43 axis in the late stages of the viral cycle points to the requirement of a stable tubulin cytoskeleton to accomplish Pr55^Gag^ assembly and viral egress [57,164]. In fact, different actions that promote acetylation of MTs, such as chemical inhibition of HDAC6, RNA interference of HDAC6 or TDP-43, or overexpression of a deacetylase-dead mutant of HDAC6, strongly favor HIV-1 replication (i.e., infectious viral production) in infected PBLs or permissive cells [57,58,75,164]. Stable MTs recruit key factors for Pr55^Gag^ transport to the plasma membrane [365], such as SOCS1, which in turn is used by Pr55^Gag^ as an MTs-anchoring protein to efficiently traffic to the cell surface [373,375]. Another experimental evidence that reinforces the importance of stable MTs for Pr55^Gag^ stability, transport, and HIV-1 replication is the fact that the MTs-associated IQGAP1 protein impedes viral egress by interacting with HIV-1 Pr55^Gag^, affecting its distribution and expression at plasma membrane-budding areas [379,380,381,382].

In this sense, the tubulin-deacetylase activity of HDAC6 inhibits HIV-1 production and infection capacity by targeting stable MTs and promoting antiviral autophagy [57,58,75,164]. HDAC6 targets Pr55^Gag^ and Vif proteins for autophagy clearance, inhibiting viral production [57,58,75]. The HIV-1 Nef protein, which is able to alter AFs [161], counteracts these antiviral actions of HDAC6 by targeting the tubulin-deacetylase enzyme for degradation [58]. Hence, by this action, HIV-1 Nef stabilizes acetylated MTs and impedes antiviral autophagy, assuring Vif and Pr55^Gag^ expression and assembly at cell surface budding areas and therefore promoting HIV-1 production and infection capacity [58]. Moreover, the implication of several factors in the antiviral autophagy mechanism makes it evident that they are key to fighting HIV-1 infection [56].

In this context and regarding the functional role of other HIV-1 accessory proteins during the viral cycle (i.e., Vpu against AFs-associated PSGL-1 [358] and AFs/membrane-associated BST2/tetherin [340,341,344,345,346,347,348,349,350] to prevent Env incorporation at viral budding sites and release of virions, respectively), some cytoskeleton associations appear to be toxic to the cell. Thus, HIV-1 Tat interacts with tubulin, leading to the alteration of MTs dynamics and the activation of a mitochondria-dependent apoptotic pathway. Tat uses Bim to facilitate this MTs-associated cell death signal [383]. Moreover, HIV-1 Vpr modulates MTs-dependent endocytic trafficking, negatively affecting phagosome biogenesis and maturation [384]. The relevance of this Tat- or Vpr-cytoskeleton interplay for HIV-1 pathogenesis is unknown. It is plausible that this Vpr-provoked cell dysfunction could be toxic for infected cells or cause difficult viral or opportunistic pathogen antigen processing and presentation to favor HIV-1 infection/replication and/or predisposing to different coinfections or superinfections and increase their severity [372,373,374,375,376,377]. In addition, HIV-1 Tat and Vpr proteins have been related to cell apoptosis by mitochondria or associated caspases, both of which directly correlate with cell apoptosis after phagolysosome disruption [383,385,386,387,391]. Moreover, it seems that the HIV-1 Rev (regulator of expression of viral proteins) factor [443] mediates the association of HIV-1 Pr55^Gag^-mRNA with cytoskeletal β-actin in perinuclear areas in the early phases of viral RNA export after transcription, where the Pr55^Gag^ protein accumulates [444]. Likewise, the nucleocapsid [364,445,446], Nef [161,447,448,449,450] and the large subunit of the HIV-1 RT [451] have been reported to interact with actin. Although these HIV-1 proteins have been rarely studied with respect to the cellular cytoskeleton, these promising results suggest that their association with actin could be used as a potential therapeutic target for the control of HIV-1 infection.

Regarding the IFs and late stages of the viral cycle, the vimentin/M2BP complex recruits the HIV-1 Pr55^Gag^ polyprotein to the vimentin IFs in an M2BP-dependent manner [426]. These associations could represent some kind of trap for HIV-1 Pr55^Gag^ that results in the inhibition of its transport to the plasma membrane and viral replication impairment [426,427]. It is important to consider that this IF protein, in its secretable form, is the object of several studies in HIV-1 patients to consider vimentin as a potential biomarker for HIV-1 infection/viremia evolution or ART efficiency and will depict new insights against HIV-1 fighting, improving early diagnosis and treatment [428].

These data may indicate that in infected cells, vimentin IFs could be targeted by the virus. In fact, vimentin, desmin, and glial fibrillary acidic IFs proteins seem to act as cleavable substrates for the HIV-1 PR enzyme [429]. The amino-terminal polypeptides of vimentin are responsible for changes in nuclear architecture associated with HIV-1 PR activity in tissue culture cells isolated from infected individuals, and condensation or degeneration of nuclear chromatin has been described in cells from a variety of tissues [433]. Furthermore, infection with HIV-1 includes an increase in the incidence of cancer, particularly lymphomas. Attempts at identifying an additional agent responsible for non-Hodgkin’s lymphoma in a large HIV-1-infected cohort have failed [434], raising the possibility that HIV-1 itself is directly responsible. In this sense, it is conceivable that HIV-1 PR-generated amino-terminal peptides from vimentin may interfere with host cell gene expression [429,435]. Likewise, this processed vimentin fragment could be important for HIV-1 to achieve the replication cycle, since PR-derived vimentin fragments disrupt vimentin IFs organization [430,431]. These perturbed vimentin IFs may not retain the Pr55^Gag^ protein, and, as occurred in cells lacking the vimentin/M2BP complex [426,427], HIV-1 PR may target vimentin IFs to ensure Pr55^Gag^ transport to the plasma membrane and subsequent Pr55^Gag^ assembly and viral particle release. Therefore, the study of vimentin and drugs targeting these IFs could be of importance to develop strategies to impede viral release and may reinforce the antiviral effect of HIV-1 PR inhibitors. Hence, the tumor inhibitor withaferin A (WFA), derived from the plant *Withania somnifer* [452], appears to aggregate vimentin IFs [453,454]. Although extracts of *Withania somnifera* roots have been found to inhibit replication of two primary isolates of HIV-1 through cell-associated and cell-free assays [455], there is no functional or molecular evidence associated with the specific targeting of vimentin IFs by these plant extracts. In fact, WFA acts on several cell substrates different from vimentin, such as signal transducer and activator of transcription 1 and 3 (STAT1 and STAT3), notch receptor 1 (Notch1), forkhead box O3 (FOXO3; previously named FOXO3A), protein kinase C (PKC), p38 mitogen-activated protein kinase (p38 MAPK), c-Jun N-terminal kinase (JNK), AKT serine/threonine kinase (Akt), and extracellular signal-regulated kinases (ERKs) (reviewed in [456]), and presents proapoptotic activity [452,457].

All these results and scientific efforts point out the future perspectives directed to control and eradicate HIV-1 [458,459,460,461,462,463,464], so that the cytoskeleton, associated modulatory receptors/ligands and/or signals could be a therapeutic target to prevent the reservoir [465], both in newly infected patients, when the reservoir has not yet settled, and in persistently infected patients. However, the modification of cytoskeletal dynamics by vinca alkaloids and taxanes [466,467], as well as HDAC histone inhibitors (HDACis) [468,469,470,471,472,473], could lead to repercussions, as has been observed in the case of cancer patients [466,467,474]. For example, HDACis are used to reduce tumor heterogeneity [475], but this could enhance HIV-1 infection, as MTs are acetylated and could facilitate pore fusion formation and infection/replication, which is why drug therapy should be carefully selected. Some anticancer drugs, such as vincristine, which binds to tubulin, stopping tubulin dimers from polymerizing, could lead to neurological complications that are demonstrated to be reduced in the presence of an HDAC6 inhibitor [476]. In this context, combination therapy with HDAC6 inhibitors additionally inhibits tumor growth in vivo. Nevertheless, some studies have demonstrated that multiple MTs modulators, such as paclitaxel, vinblastine, colchicine, and nocodazole, do not seem to disturb MTs network in transformed and resting CD4+ T cells [209]. Although these drugs disrupted MTs integrity, almost no inhibition of HIV-1 infection was observed, but this could be due to the possible reassembly of the MTs network following virus and drug removal, as MTs-related molecules showed the ability to inhibit HIV-1 infection [57,71,75,164,166]. These are a good example of the frail cell balance that should be pursued when a treatment against HIV-1 at this level is wanted. Therefore, it is necessary to implement good strategies to delve into the regulatory mechanisms of the cytoskeleton during HIV-1 infection to prevent the entry and establishment of viral reservoirs but also, once the virus has been integrated into the target cell genome, to impede viral assembly, and budding and adequate maturation.

Furthermore, some drugs are toxic to the cell, as discussed for HDAC inhibitors that are implied directly into MTs activity, so better designed drugs that could specifically act on MTs regulators are very attractive. It is important to point out that targeting specific molecules in the cell may have the benefit of avoiding the emergence of new resistances. A good example of all this is a recent study of human cytomegalovirus (hCMV or human herpesvirus 5 (HHV-5)) infection that shows that a developed myristoylated peptide that targets EB3 (MT plus-end-binding protein 3) blocked EB3-mediated enrichment of MTs regulatory proteins and suppressed acetylation of MTs, nuclear rotation, and infection [477].

Understanding the molecular mechanisms behind cytoskeletal-HIV-1 interactions may provide new insights into cargo trafficking across the cytoplasm and a specific track of how HIV-1 moves during its life cycle and exploits cytoskeletal machinery. It is clear that HIV-1 regulates and interacts with cytoskeletal structures and components throughout its life cycle and is associated with viral pathology (reviewed in [68,69,119,130,166,478,479,480,481,482,483,484,485,486,487,488,489,490,491,492,493,494,495,496,497,498]). This fact reasonably justifies the development of antiviral strategies based on drugs that target the cytoskeleton to block infection and viral cytotoxicity at some point of the cellular scaffold (i.e., AFs, Ifs, and MTs), as discussed above in this section and, for example, reported for drugs that disrupt the actin cytoskeleton thus impairing HIV-1 infection [120,123,162,465,499,500]. Drugs targeting MT dynamics are a choice for the treatment of cancer patients, despite their toxicity [501,502,503,504,505,506,507,508,509,510,511], and are well tolerated and successfully used in cancer patients living with HIV-1 and under ART [512,513,514,515,516,517,518,519]. Therefore, regarding these cancer-based therapeutic strategies, it is conceivable to develop cytoskeleton-based treatments to control HIV-1 infection, representing a therapeutic hope in the fight to eradicate HIV-1.

Altogether, these data indicate that knowledge of the precise mechanisms by which HIV-1 regulates and shapes the cytoskeleton and associated factors is important for understanding how HIV-1 infects and establishes viral reservoirs in the different target cells and tissues of patients to further assure viral persistence and evolution, which favors immune escape and pathogenesis, and compromises the effectiveness of ARTs [458,459,460,461,462,463,464]. In this sense, the functional characterization of the HIV-1 Env has permitted us to correlate the capacity of this viral complex to signal and tame actin and tubulin cytoskeletons to generate a cell permissive state for HIV-1 infection and, more importantly, to associate functional HIV-1 Envs with progression of the infection and pathogenicity in viremic and progressor HIV-1 patients. In contrast, defective HIV-1 Envs are associated with the natural control of the infection in LTNP-EC individuals [56,165,328]. Understanding the HIV-1/cytoskeleton interaction could hold the key to developing new therapeutic strategies to control HIV-1 infection and replication, thus paving the way to eradicate the virus from the organism.

## Figures and Tables

**Figure 1 ijms-24-13104-f001:**
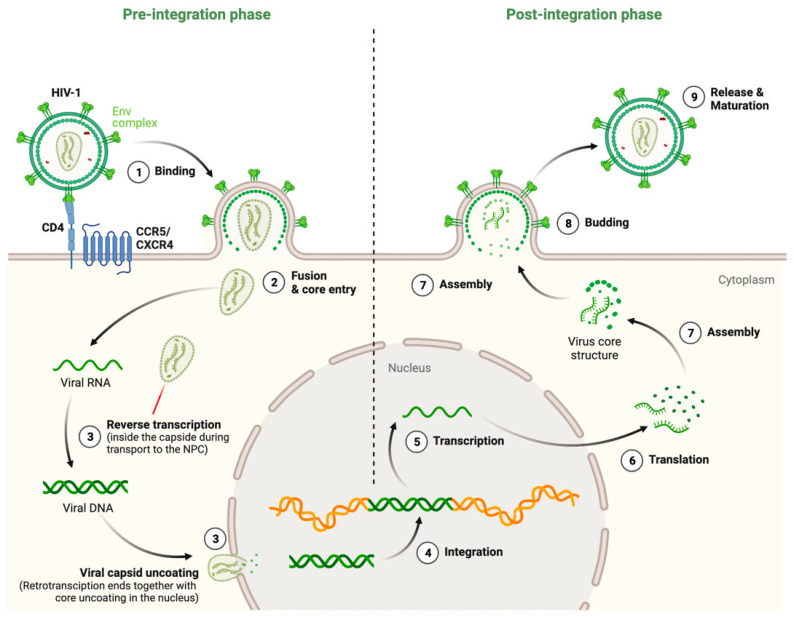
Scheme illustrating the main stages of the HIV-1 life cycle with the most representative events occurring in the pre-integration and post-integration phases of the viral infection process. HIV-1 Env/CD4-CCR5 or -CXCR4 interaction initiates the cycle driving the first steps to form the fusion pore that permits the cell entry of the viral core. The HIV-1 capsid travels to the nuclear envelope, protecting viral RNA+ and core proteins from recognition by intracellular sensors or restriction factors. Nuclear pore complexes (NPCs) receive the viral core to direct its translocation to the nucleus. The process of viral RNA+ retrotranscription to generate proviral cDNA occurs inside the viral core and is thought to be completed in the nucleus while the capsid uncoats. The viral genome is then integrated in cell host chromosomes, thus generating a cell reservoir for the virus that could be further transcribed and translated to generate RNA+ copies and all the arsenal of HIV-1 proteins necessary to transport to the plasma membrane where the virus assembles and egresses to mature and spread the infection. Designs and templates were created with BioRender.

**Figure 2 ijms-24-13104-f002:**
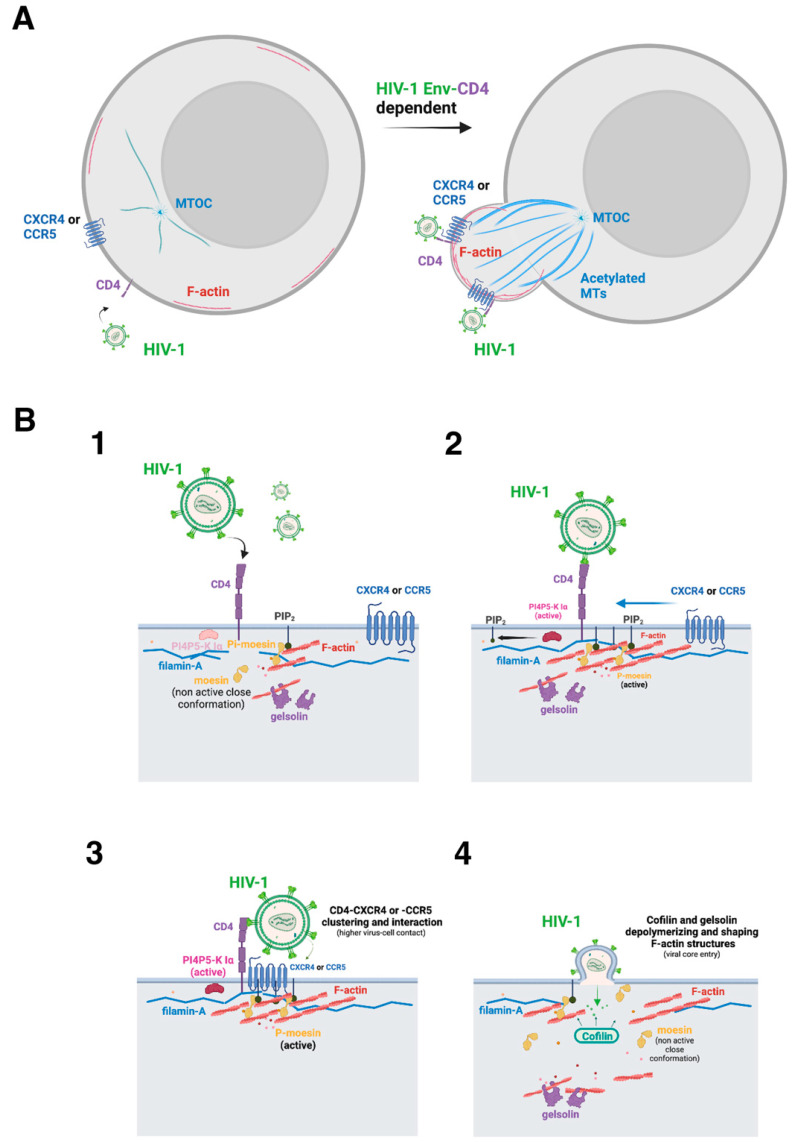
Scheme illustrating the events associated with cell actin cytoskeleton reorganization that underlie HIV-1 Env function to drive pore fusion formation, viral entry, and infection. (**A**) Functional HIV-1 Env binds to lymphocyte CD4 to signal activating actin-associated factors that reorganize cortical microfilaments to create a pseudopod “hot zone” where viral receptors aggregate and the fusion pore is formed, thus initiating the infection process, as represented in the next panel. (**B**) The illustration represents the main molecular events reported to be involved in HIV-1 Env-mediated pore fusion formation, viral entry, and infection that occurred at the pseudopod structure (panel (**A**)). The first HIV-1 Env/CD4 interaction (1), through the viral gp120 subunit, activates the cytoplasmic dormant actin-adaptor moesin, which is phosphorylated in its actin-binding C-terminal domain, thus favoring F-actin polymerization and anchoring cortical AFs to the inner leaflet of the plasma membrane either directly by its association with PIP_2_ or through several receptors (2). HIV-1 Env/CD4 binding also activates the PI4P5-K Iα kinase that produces PIP_2_ (2). This PIP_2_ molecule could mediate the plasma membrane/moesin/AFs link by recruiting the N-terminal domain of moesin (2 and 3). These events polymerize and reorganize the actin cytoskeleton around the virus–cell contact regions (3). Filamin-A stabilizes these actin areas by anchoring AFs with the cytoplasmic domains of the HIV-1 CD4, CXCR4, and CCR5 receptors (3). Therefore, filamin-A and moesin concentrate viral receptors in F-actin capping areas. These cortical AFs should be shaped in size by the actin-severing protein gelsolin to be reorganized and alter cell surface dynamics (4), generating a pseudopod region (see panel (**A**)) where the actin and tubulin cytoskeleton, their associated factors and viral receptors, aggregate and interact. It is thought that this capping region or “hot zone” generated by HIV-1 Env increases the probability of the virus recognizing its receptors and promoting the formation of the fusion pore (i.e., membrane–lipid exchange) by linking cortical AFs with the cell surface at regions where HIV-1 Env anchors (i.e., through the gp41 fusion protein). The actin-disrupting cofilin factor depolymerizes the cortical node created to form the fusion pore to permit the HIV-1 capsid to enter cells and encounter the tubulin cytoskeleton to travel to the nucleus (4). This step is also favored by the dephosphorylation of the viral activated moesin that releases AFs from their attachment to the plasma membrane (4). Designs and templates were created with BioRender.

**Figure 3 ijms-24-13104-f003:**
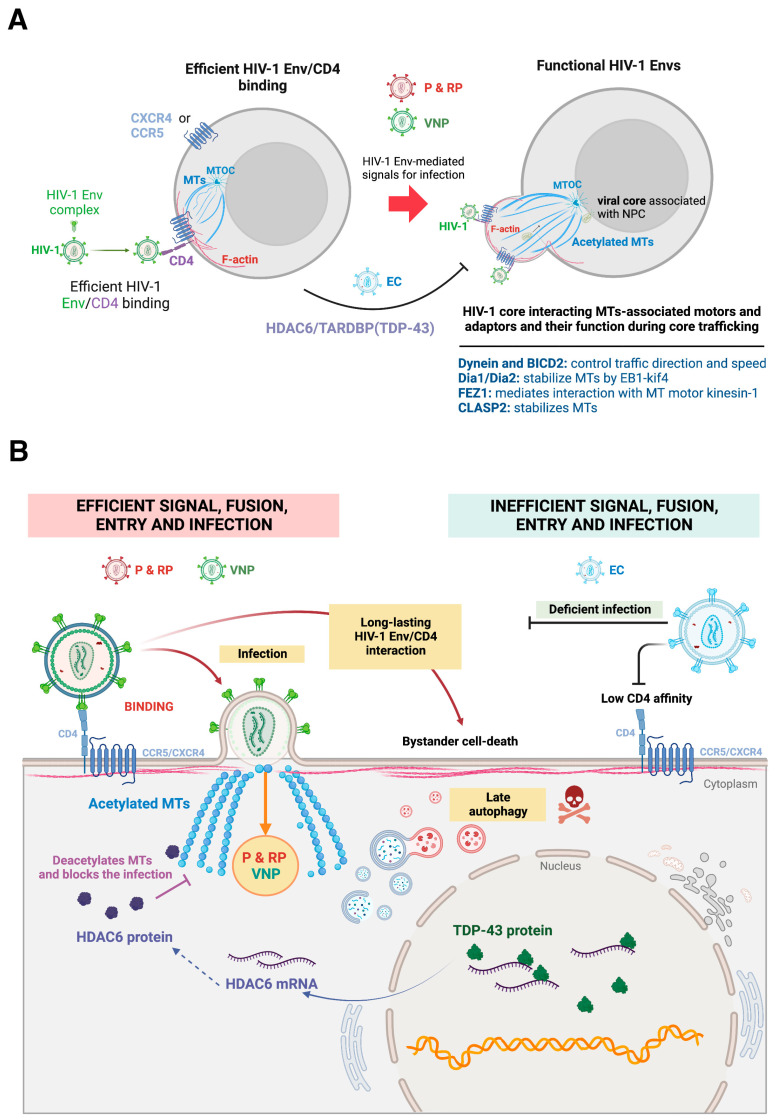
Scheme illustrating the events associated with tubulin cytoskeleton reorganization and modification that underlie HIV-1 Env function to drive pore fusion formation, viral entry, and infection. (**A**) Functional HIV-1 Envs (P, RP and VNP) signal through CD4 to stabilize MTs by acetylation. Acetylated MTs project to the virus–cell contact regions near the plasma membrane in AFs-associated pseudopods, facilitating HIV-1 Env (gp41) fusion activity and viral entry. This HIV-1-mediated MTs reorganization and PTM are under the control of HDAC6 tubulin-deacetylase and the HDAC6/TARDBP (TDP-43) axis, which inhibit viral infection. Notably, functional HIV-1 Envs from viruses of patients with viremic and progressors clinical outcomes overcome the antiviral activity of the tubulin-deacetylase HDAC6 (i.e., the HDAC6/TARDBP axis), thereby stabilizing MTs to productively infect cells. In contrast, deficient HIV-1 Envs from LTNP-EC individuals could not acetylate MTs and were not capable of infecting cells. Once the viral core enters the cell, it uses stable MTs, associated adaptors, and molecular motors to travel to the nuclear envelope, such as dynein and BICD2 (control of HIV-1 capsid transport direction and speed), Dia1/Dia2 (stabilize MTs by acting on the EB1-kif4 pathway to facilitate perinuclear trafficking), FEZ1 (associates viral core with kinesin-1 to reach the nucleus following tubulin tracks), and/or CLASP2 (EB1-associated +TIPs protein that stabilizes MTs and recruits viral core). In the nuclear envelope, HIV-1 capsid interacts with nucleoporins associated with NPCs to translocate into the nucleus. (**B**) Functional HIV-1 Envs from P, RP, and VNP patients trigger MT acetylation in a CD4-dependent manner to promote pore fusion and evade the antiviral action of endogenous tubulin-deacetylase HDAC6, which is under the control of TARDBP (TDP-43). These functional Envs could also promote late autophagy by long-lasting contact with CD4, which provokes cell death in noninfected bystander cells. In this sense, HIV-1 Envs from viruses of LTNP-EC individuals are insufficient to form the fusion pore and infect due to their inability to escape the tubulin-deacetylase barrier. Notably, viral particles bearing EC-Envs gain infection function when HDAC6 or the HDAC6/TARDBP axis is inhibited by either using HDAC6 inhibitors or its nonactive mutants or interfering with HDAC6/TARDBP mRNA, thereby stabilizing MTs to generate a cell permissive state for infection. Designs and templates were created with BioRender.

**Figure 4 ijms-24-13104-f004:**
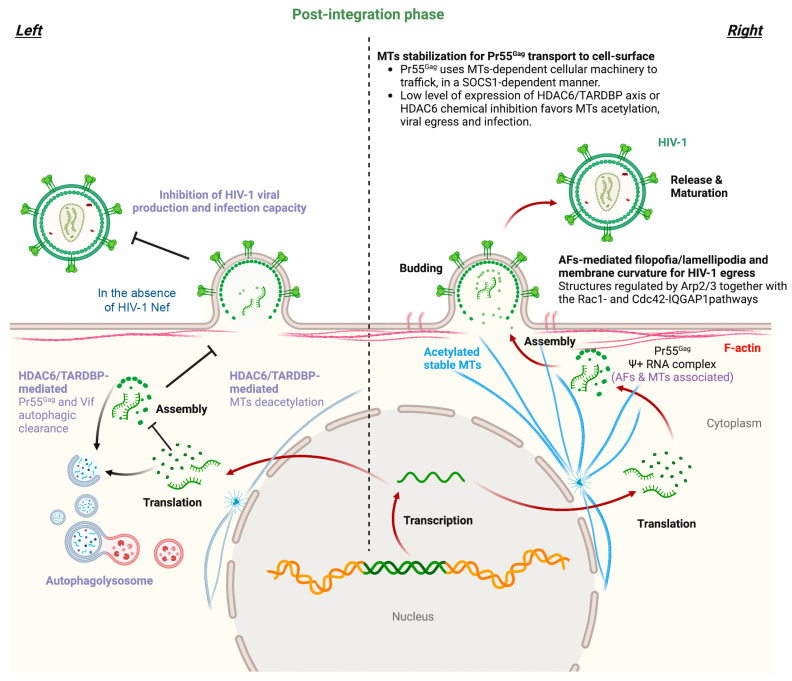
Scheme illustrating the late stages of the HIV-1 life cycle and associated tubulin and actin cytoskeleton dynamics. The transcription and translation of the integrated HIV-1 genome generate viral RNA+ and proteins, with the structural Pr55^Gag^ polyprotein being relevant to recognize viral RNA+ and recruit a complex to stable MTs to travel to the plasma membrane (**right panel**). In fact, HIV-1 Pr55^Gag^ uses MTs-dependent cellular machinery to traffic to the cell surface in a SOCS1-dependent manner. Pr55^Gag^ also interacts with F-actin, thus assembling at the cell surface in areas where AFs are remodeled by the Arp2/3- and Rac1/Cdc42/IQGAP1 pathways, inducing F-actin filopodia and lamellipodia structures that finally generate high positive membrane curvatures where HIV-1 egresses (**right panel**). The MT-associated HDAC6 enzyme deacetylates stable MTs, thereby impeding Pr55^Gag^ cell surface localization (**left panel**). Likewise, HDAC6 tubulin-deacetylase function is required to target HIV-1 Pr55^Gag^ and Vif proteins for autophagy degradation, thus inhibiting viral production and infection capacity (**left panel**). HIV-1 Nef counteracts these antiviral actions by targeting the MT-associated enzyme, thereby stabilizing MTs and the HIV-1 Pr55^Gag^ and Vif proteins. Therefore, HIV-1 Nef promotes the transport of Pr55^Gag^ to the cell surface where it assembles to form viral particles that incorporate Vif to ensure the infectivity of HIV-1. Designs and templates were created with BioRender.

**Table 1 ijms-24-13104-t001:** Cellular and viral factors that influence actin cytoskeleton and associated membrane dynamics events during HIV-1 early infection.

Cellular Factor	Cell Function	Impact on HIV-1 Infection ^1^
Moesin	Activated by HIV-1 Env/CD4 interaction to polymerize F-actin and link these polymerized AFs with plasma membrane, thus driving pseudopod formation, CD4-CXCR4 aggregation and direct interaction. Favors plasma membrane dynamics to form fusion pore.	+
Filamin-A	Aggregates with polymerized AFs to link them with cytoplasmic regions of HIV-1 CD4 and CXCR4 receptors, thereby anchoring cell surface with AFs. This process promotes CD4-CXCR4 aggregation.	+
PI4P5-K Iα	Promotes HIV-1 Env/CD4-mediated PIP_2_ production and enhances viral fusion and infection by regulating plasma membrane fluidity and activation of actin-binding proteins.	+
Arf6	GTPase required for HIV-1 Env-mediated pore fusion formation, viral entry, and infection by assuring cell surface regeneration by trafficking Arf6/PIP_2_-vesicles.	+
Gelsolin	Regulates productive viral infection by cleaving HIV-1 Env/CD4-promoted AFs into short dynamics filaments that drive actin cytoskeleton reorganization and pseudopod formation where CD4/CXCR4-CCR5 aggregates. Unbalanced gelsolin expression levels impair the above-described processes leading to the inhibition of the infection.	+/−
Cofilin	Depolymerizes AFs at postfusion steps to allow viral capsid entry into the cell.	+
EWI-2 and α-actinin	Their silencing has been associated with T-cell infection. Unknown mechanism of action at prefusion or postfusion steps.	−
Dia1/Dia2	Facilitate the intracellular motility of the viral capsid and Dia2 could bind viral capsid to mediate the uncoating process and facilitate retrotranscription at postfusion steps. This process could be detrimental to the infection if it exposes the viral genome before entry the nucleus of cells.	−/+
**Viral factor**	**Function of the viral factor**	**Impact on HIV-1 infection ^1^**
Nef	Disrupting cortical AFs at postfusion steps to allow capsid entry?	+
Nef	Enhancing retrotranscription activity by associating the PIC complex with cortical actin cytoskeleton at postfusion steps.	+

^1^ (+) represents the promotion of HIV-1 infection. (−) represents inhibition of HIV-1 infection.

**Table 2 ijms-24-13104-t002:** Cellular and viral factors that influence MT stability and dynamics during early HIV-1 infection.

MTs-Associated Cell Factor	Function of the Cellular Factor	Impact on HIV-1 Infection ^1^
HDAC6	Deacetylates MTs blocking HIV-1 Env/CD4-mediated cell signal and subsequent pore fusion formation, viral entry, and infection.	−
TDP-43	Regulates mRNA and protein levels of the tubulin-deacetylase HDAC6 to control cell permissivity to HIV-1 infection.	−
Dynein and BICD2	MTs molecular motors that control HIV-1 capsid travel speed and transport direction, to traffic to the NPC.	+
Dia1/Dia2	MTs stabilization and remodeling by acting on the EB1-kif4 pathway during infection to facilitate perinuclear trafficking of HIV-1 cores.	+
FEZ1	Mediates the association of the HIV-1 core with the MT motor kinesin-1 to reach the nucleus, a process regulated by MARK2.	+
CLASP2	An EB1-associated +TIPs protein regulating cortical capture and MT stabilization that binds HIV-1 capsids.	+
LC3	Interacts directly or indirectly with MTs during autophagy process.	−
**Viral factor**	**Function of the viral factor**	**Impact on HIV-1 infection ^1^**
Capsid	Associates with MTs adaptors/stabilizers and molecular motors to traffic to the nucleus.	+

^1^ (+) represents the promotion of HIV-1 infection. (−) represents inhibition of HIV-1 infection.

**Table 3 ijms-24-13104-t003:** Cellular and viral factors that influence actin cytoskeletal dynamics during the late stages of the HIV-1 viral cycle.

Cell Factor	Function of the Cellular Factor	Impact on HIV-1 Infection ^1^
ERM	Secure efficient virus spread not only by enhancing virion infectivity but also by preventing excessive membrane fusion at the virological synapse (VS).	+
EWI-2/CD81	Difficult HIV-1 Env-mediates cell-to-cell fusion.	−
BST2/tetherin	Links AFs with plasma membrane where it sequesters the viral Env and preventing the virus escape from cell.	−
PSGL-1	Inhibits RT in target cells and diminished the infectivity of released virions.	−
Arp2/3 and Rac-1/Cdc42- IQGAP1 pathways	Remodeling of cortical AFs in areas of high positive membrane curvature, within Arp2/3- and Rac1/Cdc42/IQGAP1-dependent F-actin filopodia and lamellipodia structures, which enables HIV-1 to bud and cell-to-cell spreading.	+
AFs	Cortical AFs bind HIV-1 Pr55^Gag^ protein by the NC domain.	+
**Viral factor**	**Function of the viral factor**	**Impact on HIV-1 infection ^1^**
Pr55^Gag^	Actin interaction facilitating viral egress.	+
Vpu	Counteracts BST-2/tetherin, EWI-2, and PSGL-1 antiviral activities.	+

^1^ (+) represents the promotion of HIV-1 replication. (−) represents inhibition of HIV-1 replication.

**Table 4 ijms-24-13104-t004:** Cellular and viral factors that influence MTs dynamics during HIV-1 viral egress and maturation.

Cell Factor	Function of the Cellular Factor	Impact on HIV-1 Infection ^1^
HDAC6	Colocalized with and deacetylates MTs to promote autophagy degradation of Vif and Pr55^Gag^ viral proteins, thus stabilizing the restriction factor A3G, forming an A3G/HDAC6 antiviral complex, an inhibiting HIV-1 viral production and infectiveness.	−
TDP-43	Regulates HDAC6 mRNA and protein levels promoting its antiviral function by tubulin-deacetylation and dependent anti-HIV-1 autophagy. Controls HIV-1 production and the infection capacity of viral particles.	−
SOCS1	Associates with MTs and allows Pr55^Gag^ efficient transport to plasma membrane.	+
IQGAP1	A MTs-associated signaling scaffold protein that impedes viral egress by interacting with HIV-1 Pr55^Gag^, affecting its distribution and expression at plasma membrane-budding areas.	−
**Viral factor**	**Function of the viral factor**	**Impact on HIV-1 infection ^1^, cell-function or viral toxicity ^2^**
Nef	Targets tubulin-deacetylase HDAC6 to stabilize MTs, HIV-1 Pr55^Gag^ and Vif proteins, assuring virus production and the infection capacity of HIV-1.	+
Pr55^Gag^	HIV-1 Pr55^Gag^ uses MTs-dependent cellular machinery to traffic to the cell-surface in a SOCS1-dependent manner.	+
Tat	Interacts with tubulin leading to the alteration of MTs dynamics and the activation of a mitochondria-dependent apoptotic pathway. Tat uses Bim to facilitate this MTs-associated cell-death signal.	*
Vpr	Modulator of the MTs-dependent endocytic trafficking, negatively affecting phagosome biogenesis and maturation.	*

^1^ (+) represents the promotion of HIV-1 replication. (−) represents inhibition of HIV-1 replication. ^2^ (*) represents alteration of cell function or viral toxicity.

## Data Availability

Not applicable.

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
