# Peer review of "HIV Infection: Shaping the Complex, Dynamic, and Interconnected Network of the Cytoskeleton"

_ijms, 2023, doi:10.3390/ijms241713104_

Round 1

Reviewer 1 Report

Paper has many old citations.

Authors provided detailed discussion about the stages of HIV-1 infection as it relates to cytoskeleton and associated factors, however, little information or recommendations were discussed for therapeutic strategies to curb/treat/battle HIV-1. Please discuss more.

Author Response

Point-by-point reply to the Reviewer # 1

("Please see also the attachment") 

Comments and Suggestions for Authors:

Paper has many old citations.

We thank Reviewer #1 for this comment on our work and highlighting the relevance of the Review in the evaluation checking list of the IJMS.

We have checked that all references are relevant to the contents of the manuscript, including some new references (mainly, Reviews about HIV-1 and cytoskeleton) in the Discussion section of the Ms (lines 1164-1186) and associated with this comment. We aimed to refer to the first works addressing the potential involvement of the cell cytoskeleton in HIV infection, although several of them were developed by using drugs that severely and nonspecifically act on cystoskeleton dynamics. These works open the door to more accurately explore the functional role of the cell cytoskeleton during HIV-1 infection. We would like to consider and maintain these original works in the Ms to accomplish an educational task for young scientific readers.

Authors provided detailed discussion about the stages of HIV-1 infection as it relates to cytoskeleton and associated factors, however, little information or recommendations were discussed for therapeutic strategies to curb/treat/battle HIV-1. Please discuss more.

We would like to thank Reviewer #1 for this suggestion. We have addressed this subject in the Discussion section. In this sense, we have now added some new references that address some potential therapeutic strategies “to curb/treat/battle HIV-1” by targeting the cell cytoskeleton and/or associated factors and discuss them. We feel this aspect is clearly remarked in the Discussion of the manuscript and highlights the importance of understanding cytoskeleton dynamics during HIV-1 infection to develop new anti-HIV-1 drugs and strategies. In this sense, we feel this aspect is clearly remarked in the Discussion of the manuscript and highlights the importance of the understanding of cytoskeleton dynamics during HIV-1 infection to develop new anti-HIV-1 drugs and strategies, as discussed in lines 1124-1186.

We have highlighted the modifications in red in the text of the Ms (ijms-2575562).

We would like to thank Reviewer #1 for all these comments that have improved our work.

Reviewer 2 Report

The authors should add a section discussing about the literature where drugs which can disrupt the cytoskeleton elements have been shown to inhibit HIV -I1 replication at add it to the figure section as well.

I have some concern about the core disassembly. It will be good if the authors provide the literature if they are correct otherwise correct the text and figure. I suppose the core disintegrates in cytoplasm and then it goes inside nucleus as PIC while as the authors show core goes to the nucleus .

NO comments

Author Response

Point-by-point reply to the Reviewer # 2

("Please see also the attachment.")

Comments and Suggestions for Authors:

The authors should add a section discussing about the literature where drugs which can disrupt the cytoskeleton elements have been shown to inhibit HIV -1 replication at add it to the figure section as well.

We thank Reviewer #2 for this comment on our work and highlighting the relevance of the Review in the evaluation checking list of the IJMS.

We hope that the Reviewer #2 understands that we have decided not to include in the Ms a specific Section to address the use of drugs that target the cytoskeleton in every reported study about the role of cytoskeleton on HIV-1 infection. This is due to the severe and irreversible effect exerted by the different drugs or toxins that could be used to target actin or tubulin cytoskeleton, for example, such as phalloïdin or jasplakinolide, or nocodazole and taxol, among other derivatives. We aimed to present and discuss data that shed light on the fine mechanisms that govern cell cytoskeleton reorganization and dynamics and that are triggered by HIV-1 to infect target cells, as well as the functional effects on the viral life cycle driven by cytoskeleton-associated factors. The use of drugs makes it difficult to understand the fine mechanisms underlying virus-mediated cytoskeleton dynamics. In this sense, we have cited some “classical references” in the Review that correspond to first works that use drugs or toxins to study the cell cytoskeleton in HIV-1 infection. These works open the door to more accurately explore the functional role of the cell cytoskeleton during HIV-1 infection.

Furthermore, and according to this remark by Reviewer #2, we have cited in the Discussion section of the manuscript some potential therapeutic strategies to battle HIV-1 by targeting the cell cytoskeleton and/or associated factors and discussing them. We prefer to highlight the importance of drugs targeting cell cytoskeleton at this level and in the Discussion section than abord all works using drugs-toxins to study HIV-1 infection in the main text of the Ms with a particular section that are out of the focus of the main idea that drives the Review about molecular mechanisms. In this sense, we feel this aspect is clearly remarked and discussed in the Discussion section of the manuscript and highlights the importance of the understanding of cytoskeleton dynamics during HIV-1 infection to develop new anti-HIV-1 drugs and strategies, as analysed in lines 1124-1186.

I have some concern about the core disassembly. It will be good if the authors provide the literature if they are correct otherwise correct the text and figure. I suppose the core disintegrates in cytoplasm and then it goes inside nucleus as PIC while as the authors show core goes to the nucleus.

We would like to thank Reviewer #2 for this comment.

We have addressed this controversial and complex issue in the understanding of the early stages of HIV-1 infection (preintegrative phase), analysing and discussing relevant works that point to the fact that the completion of viral genome reverse transcription and viral core uncoating occur in the nucleus rather than in the cytoplasm. In this context, we have considered the interaction of the viral core with the cell cytoskeleton and associated factors (e.g., molecular motors) in the preintegration phase of the viral cycle and discussed the works that have studied the importance of the interplay of the Pr55Gag-RNA+ complex with the cell cytoskeleton for the postintegrative phase of the infection cycle.

In this sense, we have added this sentence to more clearly emphasize this issue in the preintegrative phase (lines 59-63; text marked in red), as follows:

“However, several studies have indicated that the cone-shaped capsid remains intact during travel to the nucleus, which encases the genome and replication proteins. Therefore, the viral core serves as a reaction container for reverse transcription, a pro-cess that seems to be completed with uncoating at the nucleus [15,16,29-44].”

We have also added associated text (in red) in lines 346-355, as follows:

“The uncoating process is still a great topic of discussion, as researchers have three different points of view depending on whether the capsid is totally uncoated in the cyto-plasm immediately during entry through the fusion pore [42,44,126-132], partially uncoated in the cytoplasm until the preintegration complex (PIC) reaches the nuclear pore [40,133-135] or uncoating occurs entirely at the nucleus [136-139]. However, as indicated above, recent studies indicated that the HIV-1 capsid remains intact during travel to the nuclear envelope, which encases the genome and replication proteins. Therefore, the viral core serves as a reaction container for reverse transcription, a process that seems to be completed with uncoating at the nucleus [15,16,29-37].”

The general analysis addressed in our Review is supported by the research works reported in the past 30 years that have seen a constant evolution of our perspective regarding HIV-1 capsid integrity and uncoating. This started in the 1990s with pioneering experiments by Haseltine and Fassati, who biochemically characterized HIV-1 RTC/PIC isolated from infected cells (Bukrinsky MI., et al. (1993) PNAS 90(13):6125-29; Farnet CM and Haseltine WA (1991) J. Virol. 65(4):1910-15; Fassati A and Goff SP (2001) J. Virol. 75(8):3626-35; Miller M., et al. (1997) J. Virol. 71(7):5382-90) and proceeded with labelling of subviral complexes by Hope and colleagues in the early 2000s (McDonald D., et al. (2002) J. Cell Biol. 159(3):441-52; Campbell EM., et al. (2008) Trends Microbiol. 16(12):580-87; Campbell EM., et al. (2007) Virology 360(2):286-93) to recent studies applying advanced microscopic methods (comprehensively reviewed in Muller, TG., et al. (2022) Annu Rev Virol. 9, 261-284 (doi:10.1146/annurev-virology-020922-110929); and in Arhel, N (2010) Retrovirology, 7, 96 (doi:10.1186/1742-4690-7-96)). Recent studies indicated that the cone-shaped capsid, which encases the genome and replication proteins, not only serves as a reaction container for reverse transcription and as a shield from innate immune sensors but also may constitute the elusive HIV-1 nuclear import factor. As we have addressed in this Review, the rupture of the viral capsid may be triggered in the nucleus by completion of reverse transcription, by yet-unknown nuclear factors, or by physical damage, and it appears to occur in close temporal and spatial association with the integration process (comprehensively reviewed in Muller, TG., et al. (2022) Annu Rev Virol. 9, 261-284 (doi:10.1146/annurev-virology-020922-110929); and in Arhel, N (2010) Retrovirology, 7, 96 (doi:10.1186/1742-4690-7-96)).

According to the use of the cell cytoskeleton by HIV-1, we feel that we have more clearly emphasized, cited and discussed the main works addressing this complex issue of uncoating and reverse transcription during the preintegrative phase of the viral life cycle.

We have highlighted all the main modifications in red and in red‒yellow (for some English corrections) in the text of the Ms (ijms-2575562).

We would like to thank Reviewer #2 for the English language remark. The English grammar has been edited and corrected by AJE (https://www.aje.com/services/digital/) through our institutional cooperation.

We would like to thank Reviewer #2 for all these comments and discussion about this Review.
